
# Effects of Marine Organic Aerosols as Sources of
# Immersion-Mode Ice Nucleating Particles on High Latitude
# Mixed-Phase Clouds
Xi Zhao[1], Xiaohong Liu[1], Susannah Burrows[2], and Yang Shi[1],
[1]Department of Atmospheric Sciences, Texas A&M University, College Station, Texas, 77840, USA
[2]Pacific Northwest National Laboratory, Richland, Washington, 99352, USA

*Correspondence to*: Xiaohong Liu (xiaohong.liu@tamu.edu)



**Abstract.** Mixed-phase clouds are frequently observed in the Arctic, Antarctic, and over
the Southern Ocean, and have important impacts on the surface energy budget and
regional climate. Marine organic aerosol (MOA), a natural source of aerosol emitted over
~70% of Earth's surface, may significantly modify the properties and radiative forcing of
mixed-phase clouds. However, the relative importance of MOA as a source of ice
nucleating particles (INPs) in comparison to mineral dust, and its effects as cloud
condensation nuclei (CCN) and INPs on mixed-phase clouds are still open questions. In
this study, we implement MOA as a new aerosol species into the Community
Atmosphere Model version 6 (CAM6), the atmosphere component of the Community
Earth System Model version 2 (CESM2), and allow the treatments of aerosol-cloud
interactions of MOA via droplet activation and ice nucleation. CAM6 reproduces
observed seasonal cycles of marine organic matter at Mace Head and Amsterdam Island
when the MOA fraction of sea spray aerosol in the model is assumed to depend on sea
spray biology, but fails when this fraction is assumed to be constant. Model results
indicate that marine INPs dominate primary ice nucleation below 400 hPa over the
Southern Ocean and Arctic boundary layer, while dust INPs are more abundant elsewhere.
By acting as CCN, MOA exerts a shortwave cloud forcing change of –2.78 W m$^{-2}$ over
the Southern Ocean in the austral summer. By acting as INPs, MOA enhances the
longwave cloud forcing by 0.35 W m$^{-2}$ over the Southern Ocean in the austral winter.
The annual global mean net cloud forcing changes due to CCN and INPs of MOA are
–0.35 and 0.016 W m$^{-2}$, respectively. These findings highlight the vital importance of
Earth System Models to consider the MOA as an important aerosol species for the
interactions of biogeochemistry, hydrological cycle, and climate change.


## 1 Introduction

Ice crystals in clouds play a critical role in determining cloud phase, lifetime, electrification, and radiative properties. As a result, cloud ice influences precipitation and cloud radiative forcing. To quantify the impact of ice crystals on the hydrologic cycle and energy budget of the Earth system, it is important to advance the process-based understanding of initiation and evolution of ice particles. Ice particles can be initialized by homogeneous freezing or by heterogeneous nucleation. Homogeneous freezing of cloud droplets and aerosol solution droplets happens when air temperature is below approximately –38˚ C. In mixed-phase clouds in which air temperature is between –38˚ C to 0˚ C, ice is initialized only by heterogeneous nucleation on ice nucleating particles (INPs) (Kanji et al., 2017).

INPs have different characteristics in their compositions and origins. Previous studies (Hoose and Möhler, 2012; Murray et al., 2012; Kanji et al., 2017) have shown that mineral dust, primary bioaerosols (e.g., fungal spores, bacteria, and pollen), and volcanic ash can be effective INPs. However, large uncertainties exist surrounding the ice nucleating properties of black carbon and organic carbon from biomass burning and fossil fuel combustion. A majority of INPs are of terrestrial origin. Due to their large emission quantities and high efficiency at forming ice, mineral dust may play a dominant role in ice formation over continents. However, in remote oceanic regions where terrestrial INPs are rare, the aerosol species contributing to INPs and the mechanisms for ice initialization remain poorly understood. Recent observational and modelling studies have shown that marine organic aerosol (MOA) is potentially an important source of INPs over remote oceanic regions (Wilson et al., 2015; DeMott et al., 2016; Vergara-Temprado et al., 2017; Huang et al., 2018; McCluskey et al., 2019).

MOA can be generated from both primary and secondary processes during ocean biological activities, producing either water-soluble or insoluble organic aerosols. Previous studies have inferred that water-insoluble marine organic matter is mainly derived from the primary emissions of sea spray aerosols (SSAs) (Ceburnis et al., 2008). In this production process, SSAs and associated organic matter are injected into the marine boundary layer when bubbles burst at the air-sea interface. Long-term measurements of seasonal variability in SSAs (O'Dowd et al., 2004; Yoon et al., 2007; Rinaldi et al., 2013)





and organic matter in remote marine air (Sciare et al., 2009) are consistent with the
hypothesis that the amount of organic matter is associated with ocean biological activity.
Laboratory experiments have also demonstrated that the presence of phytoplankton blooms
can be associated with significant changes in the number flux and size distribution of
emitted SSAs (Alpert et al., 2015; Rastelli et al., 2017; Forestieri et al., 2018; Christiansen
et al., 2019), as well as the SSA organic content (Facchini et al., 2008; Ault et al., 2013).

Parameterizations for the primary emission of MOA have been developed with the

intention to be used in models. Most of these parameterizations relate MOA emission flux
to ocean chlorophyll a concentration [Chl-a]. An advantage of this approach is that [Chl-a]
is globally available from satellite-based measurements, especially over the remote oceans
where ground-based observations are difficult to conduct. Although [Chl-a] makes up only
a minor fraction of the organic matter in the ocean (Gardner et al., 2006), it has a long
history as a widely-used proxy for the biomass of phytoplankton in ocean surface waters
(Steele et al., 1962; Cullen et al., 1982), and has been used to derive empirical relationships
between satellite-observed [Chl-a] and the observed MOA contribution to submicron SSAs.
Several studies have also found that measured organic matter in SSA correlates more
strongly with ocean [Chl-a] than with other satellite-retrieved ocean chemistry variables,
such as particulate organic carbon, dissolved organic carbon, and colored dissolved and
detrital organic matter (O'Dowd et al., 2004; Sciare et al., 2009; Gantt et al., 2011; Rinaldi
et al., 2013).

O'Dowd et al. (2008) proposed a MOA emission parameterization, which was further

modified by Langmann et al. (2008) and Vignati et al. (2010). In this parameterization, the
fraction of emitted organic matter in SSA has a linear relationship with ocean [Chl-a] and is
not dependent on surface wind speed. Gantt et al. (2011) took a step further, and developed
an emission parameterization in which the organic matter fraction is an empirical function
of ocean [Chl-a], 10 m wind speed, and aerosol size. Both parameterizations from Gantt et
al.(2011) and Vignati et al. (2010) were found to capture the magnitude of MOA
concentrations compared to observations, but the parameterization from Gantt et al. (2011)
had a better representation of seasonal variability of MOA concentrations at Amsterdam
Island and Mace Head, Ireland (Meskhidze et al., 2011). Rinaldi et al. (2013) also
developed a MOA emission parameterization which depends on surface wind speed and





[Chl-a], and by assuming an 8–10 day time lag between upwind ocean [Chl-a] and
enhanced production of MOA the correlation between enriched MOA and [Chl-a] was
improved. Burrows et al. (2014) proposed a physically-based approach to represent MOA
emission process (i.e., OCEANFILMS) instead of using the empirical [Chl-a]. This
method was implemented in the DOE Energy Exascale Earth System Model version 1
(E3SMv1) (Golaz et al., 2019; Rasch et al., 2019), and the CCN effect of MOA on cloud
droplet activation was investigated (Burrows et al., 2018).
Recent observational evidence continuously shows the importance of MOA as INPs
in natural clouds (Wilson et al., 2015; DeMott et al., 2016; McCluskey et al., 2018a, b).
However, there have been very limited modeling studies to quantify the effects of MOA
INPs on clouds. Yun and Penner (2013) conducted the first global study of MOA on ice
formation and radiative forcing using the CAM3 model. Their study indicated that MOA
INPs are the dominant INPs for mixed-phase clouds over the Southern Hemisphere (SH),
and after including MOA INPs, the model generated a more reasonable ice water path
(IWP) compared with the International Satellite Cloud Climatology Project (ISCCP)
observation data. In their study, the model simulated frozen fraction of MOA at $-15°C$ is
3.75% for their lowest size bin (0.05 – 0.63 μm) and 100% for their larger size bins. These
values may be too high compared with both historical and recent measurements of the ice
nucleation efficiency of sea surface material (Schnell and Vali, 1975; Wilson et al., 2015)
and SSAs (DeMott et al., 2016; McCluskey et al., 2018b).
With more measurements of MOA and sea spray INPs becoming available, recent
modeling studies have been able to improve upon past MOA INP parameterizations.
Huang et al. (2018) used the ECHAM6-HAM2 model to study the MOA influence on ice
formation and climate. They followed the [Chl-a]-based of Rinaldi et al. (2013) to
represent the MOA emission and compared two empirical methods for calculating the
MOA INP efficiency (Wilson et al., 2015; DeMott et al., 2016). They found that MOA
influenced the cloud ice number concentration and effective radius only slightly, and MOA
did not exert a significant influence on the global radiative balance due to compensating
cloud responses. However, these conclusions also depend on the sensitivity of their model
to the change in INP number concentration.





In contrast to the findings of Huang et al. (2018), Vergara-Temprado et al. (2017) and
McCluskey et al. (2019) found that MOA was the dominant source of INPs over the
Southern Ocean. Vergara-Temprado et al. (2017) used the Global Model of Aerosol
Processes (GLOMAP) to investigate the relative importance of feldspar and MOA for ice
nucleation. Ice nucleation by MOA follows the Wilson et al. (2015) parameterization. This
study also found that on 10–30 % of days in the study period there were more MOA INPs
than feldspar INPs over the Northern Hemisphere (NH) Ocean. McCluskey et al. (2019)
used the aerosol concentrations calculated offline from the Community Atmosphere Model
version 5 (CAM5) to show that MOA is the dominant INPs over the Southern Ocean. Ice
nucleation by MOA follows the McCluskey et al. (2018b) parameterization.
Isolating the INP effect of MOA on clouds and radiative forcing has rarely been
examined directly, which motivates our study to address MOA ice nucleation process and
to better understand the climate influence of MOA INPs. Our approach is different from
previous studies. For example, we use a more physically-based approach (Burrows et al.,
2014) to represent MOA emission instead of the empirical [Chl-a] based method used in
Huang et al. (2018). Instead of the offline evaluation of INP parameterizations in CAM5
(McCluskey et al., 2019), this study implements the MOA emission and other process
representations in the Community Atmosphere Model version 6 (CAM6), the latest
atmosphere component of Community Earth System Model version 2 (CESM2), and
allows for the impacts of MOA on modeled clouds and radiative forcing interactively.
Lastly, we isolate the INP effect from the CCN effect of MOA in order to better understand
the MOA influence on clouds via these two mechanisms.
This paper is organized as follows. Section 2 presents the model, parameterizations of
MOA as well as model experiments. Section 3 describes the model results and comparison
with observations. Section 4 discusses the remaining questions. Section 5 summarizes and
draws the conclusions of this study.



## 2 Methods

### 2.1 Model and parameterizations

CAM6 with the Finite-Volume (FV) dynamical core (Lin and Rood, 1997) is used in this study. CAM6 treats important physical processes in the atmosphere, including radiative transfer, deep convection, cloud macrophysics, cloud microphysics, shallow convection, and planetary boundary layer turbulence. Cloud and aerosol interactions with longwave and shortwave radiation transfer are treated by the Rapid Radiative Transfer Model for GCMs (RRTMG) scheme (Iacono et al., 2008; Mlawer et al., 1997). A double-moment scheme (Gettelman et al., 2015) is used to describe the microphysical processes of cloud and precipitation hydrometeors in large-scale stratiform clouds, while the deep convection is represented by the Zhang and McFarlane (1995) scheme. CAM6 uses the Cloud Layers Unified By Binormals (CLUBB) scheme (Golaz et al., 2002; Larson et al., 2002) to unify the representations of cloud macrophysics, turbulence, and shallow convection.

The four-mode version of the Modal Aerosol Module (MAM4), which is an extension of the three-mode version of MAM (Liu et al., 2012), is used to describe the aerosol properties and processes in CAM6 (Liu et al., 2016). MAM4 uses the modal method to represent the size distributions of four aerosol modes: Aitken, accumulation, coarse, and primary carbon. The original MAM4 encompasses six aerosol species: black carbon, dust, primary organic aerosol, sea salt, secondary organic aerosol, and sulfate (Table 1). The primary organic aerosol here refers to non-marine sources of organic matter, usually from terrestrial biomass burning, fossil fuel, and biofuel burning. Aerosol species are internally-mixed within a mode and externally-mixed between modes. The mass mixing ratio of each aerosol species within a mode and the total number mixing ratio of aerosols in that mode are predicted in the model. Then the log-normal size distribution can be determined for each mode based on a prescribed geometric standard deviation (Table 1). Different aerosol species are characterized by a variety of properties such as hygroscopicity, density, and optical properties (Table 2).

While anthropogenic aerosol and precursor gas emissions are prescribed for model simulations, emissions of natural aerosols (e.g., SSA, dust) are calculated



interactively in the model. SSA in MAM is emitted following the parameterization of
Mårtensson et al. (2003) for dry particle diameters from 0.020 to 2.8 µm, and Monahan et
al. (1986) from 2.8 to 10 µm. The Mårtensson et al. parameterization is derived from
laboratory experiments in which particles were produced by bubble bursting using a
sintered glass filter in synthetic seawater. The emission rate depends linearly on the sea
surface temperature and is proportional to 10-m wind speed, raised to the power of 3.41
(Monahan et al., 1986; Gong et al., 1997).

## 2.2 MOA in CAM6

In this study, several modifications are implemented in CAM6 in order to
explicitly quantify the influence of marine organic matter on aerosols, clouds, and
radiation. These modifications are comprised of (1) emission schemes of MOA, as
introduced in section 2.2.1, and (2) ice nucleation parameterizations for MOA, as
introduced in section 2.2.2.

### 2.2.1 Emission of MOA

Three different methods for online MOA emissions are implemented in CAM6.
These methods parameterize the organic mass fraction of sea spray and use the fraction to
compute MOA emissions based on the emission rate of SSA.
The mass fraction of MOA in total SSA, $F_{MOA/SSA}$ is defined as the following:
$$F_{MOA/SSA} = \frac{M_{MOA}}{M_{sea\ spray}} = \frac{M_{MOA}}{M_{MOA} + M_{sea\ salt}} \qquad (1)$$
in which $M_{MOA}$ is the mass mixing ratio of MOA, and $M_{sea\ salt}$ is the mass mixing
ratio of sea salt. Thus, the emitted MOA mass mixing ratio can be computed as:
$$M_{MOA} = \frac{F_{MOA/SSA} \times M_{sea\ salt}}{1 - F_{MOA/SSA}} \qquad (2)$$
The emitted MOA number mixing ratio is calculated based on the emitted mass
mixing ratio and particle density of MOA, the latter of which is set to be 1601 kg m$^{-3}$
(Liu et al., 2012), as given in Table 2.
Differences between the three emission methods lie in how to determine the
organic mass fraction $F_{MOA/SSA}$. These methods are compared in this study: the first is





the Langmuir isotherm-based parameterization by Burrows et al. (2014) (B14), the
second is based on wind speed and [Chl-a] by Gantt et al. (2011) (G11), and the third,
which represents a null hypothesis, assumes a fixed mass fraction between organic matter
and sea salt (NULL).
**a. G11 emission scheme**
A chlorophyll-based emission scheme of MOA was derived based on the [Chl-a]
and the 10-m wind speed (Gantt et al. (2011), hereafter referred to as G11). In this
method, the organic mass fraction of sea spray is parameterized as:
$$F_{MOA/SSA} = \frac{\frac{1}{1+0.03\times e^{6.81\times D_p}}+0.03}{1+e^{-2.63\times(Chl-a)+0.18U_{10}}} \tag{3}$$

where $D_p$ is the dry diameter of particles.
**b. B14 emission scheme**
Different from the earlier empirical chlorophyll-based scheme, a physically-based
scheme, named OCEANFILMS was proposed for modeling the relationship between
emitted SSA chemistry and ocean biogeochemistry (Burrows et al. (2014), hereafter
referred to as B14). The Langmuir isotherm-based mechanism is adopted to describe the
organic enrichment on the bubble film. When the bubble film bursts, the film breaks up
into film drops, which are suspended in the air. After evaporation of water from these
droplets, the remaining suspending materials form MOA and sea salt aerosol particles. In
this method, the organic matter on one side of the bubble film (per area) is determined
by:
$$M_{s\_MOA} = S_m \times \theta \tag{4}$$

where $S_m$ is the organic mass per area at saturation (Table 3), and $\theta$ is the surface
coverage fraction of organics calculated based on the Langmuir adsorption equilibrium
assumption:
$$\theta = \frac{\alpha \times C_M}{1+\alpha \times C_M} \tag{5}$$

where $\alpha$ is the Langmuir parameter as prescribed in Table 3, and $C_M$ is the mass
concentration of organic matters in the ocean. $C_M$ is prescribed from the monthly mean
surface distribution of macromolecule concentrations, which is generated by ocean





biogeochemical simulations (Burrows et al., 2014). In this method, three different organic
classes are considered with molecular weights and mass per area at saturation as
prescribed in Table 3.

Based on Equations (1), (4), and (5), the organic mass fraction of sea spray is

expressed as:
$$F_{MOA/SSA} = \frac{S_m \times \frac{\alpha \times C_M}{1+\alpha \times C_M}}{S_m \times \frac{\alpha \times C_M}{1+\alpha \times C_M} + M_{s\_sea\,salt}}$$
(6)

$M_{s\_sea\,salt}$ is the sea salt mass per area of bubble surface, which is set to be 0.0035875 g
m$^{-2}$.

**c.  NULL emission hypothesis**

Null hypothesis assumes that the organic mass fraction of SSA is constant, and

does not vary geographically or seasonally. If we are to adopt a parameterization for the
seasonal dependence of MOA, it is desirable to demonstrate that the agreement with
observations of MOA is improved by such a parameterization, compared with the null
hypothesis that no such relationship exists. The choice of the "null" hypothesis is
motivated in part by Quinn et al. (2014) and Bates et al. (2020), who measured roughly
constant values of $F_{MOA/SSA}$ in SSAs generated at sea by using a floating device to
generate and sample spray, during five sea-going ship campaigns. These studies
measured $F_{MOA/SSA}$ values of roughly 0.7–0.9 in sub-0.180 µm particles, and roughly
0.05–0.3 in sub-1.1 µm particles.

Loosely following the results of Quinn et al. (2014) and Bates et al. (2020), we set

$F_{MOA/SSA}$ to 0.8 in the Aitken mode, and to 0.05 in the accumulation mode (see Table 1
for the size ranges of Aitken and accumulation modes). For comparison, Facchini et al.
(2008) measured SSA generated from oceanic water for its organic and salt content, and
found that organic matter comprised roughly 75% of particles in the size range
0.125–0.250 µm, and that this fraction decreased with increasing particle size to about 5%
of 1 µm particles. Similarly, Prather et al. (2013) analyzed sea spray generated in a wave
tank during a mesocosm bloom experiment and reported that about 80% of 0.080 µm
particles were classified as organic carbon by transmission electron microscopy (TEM)





with energy-dispersive X-ray (EDX), while a few percents of 1 μm particles were
classified as either organic carbon or biological species by the aerosol TOF mass
spectrometry (ATOFMS).

**2.2.2    Effects of MOA on clouds as CCN and INPs**

MOA is emitted into different aerosol modes depending on mixing state of MOA

and sea salt (Burrows et al., 2014, 2018). In the internally-mixed emission approach,
MOA is emitted into the accumulation and Aitken modes along with sea salt, as shown in
Table 1. In contrast, MOA is emitted into the accumulation and primary carbon modes in
the externally-mixed emission approach. Furthermore, the emission of MOA can replace
or be added to sea salt emission in terms of mass and number in the model. Burrows et al.
(2018) found that simulated MOA amounts, seasonal cycles, and impacts on CCN over
the Southern Ocean show better agreement with observations under the assumption that
emitted MOA is added to, and internally mixed with, sea salt. As shown in Table 2, the
hygroscopicity of MOA is set to be 0.1 following Burrows et al. (2014, 2018), compared
to 1.16 for sea salt. The mode hygroscopicity is calculated as the volume-weighted
average of all species in a mode, which is then used in the Abdul-Razzak and Ghan (2000)
droplet activation parameterization in CAM6.

In this study, in addition to the CCN effect of MOA, we also include its effect on

clouds as INPs. For this purpose, two different ice nucleation parameterizations for MOA
are implemented in CAM6. Additionally, we examine the relative importance of MOA to
dust INPs with different ice nucleation parameterizations.
**a.    W15 ice nucleation scheme of MOA**

An INP parameterization for MOA was proposed based on immersion-freezing

measurements of materials aerosolized from sea surface microlayer (SML) water samples
collected in the North Atlantic and Arctic Oceans (Wilson et al., 2015). In this
parameterization (hereafter as W15), the number concentration of MOA INPs is a
function of temperature ($T$) and the total organic carbon (TOC) mass concentration, given
as:





$$N_{IN,T} = TOC \times e^{(11.2186-(0.4459\times T))} \qquad (7)$$

W15 assumes that relationship between TOC and INPs in airborne sea spray is the
same as that in SML samples due to limited measurement data in the early stage.
However, recent research suggests that INPs may be transferred differently from TOC
during the sea spray production (Wang et al., 2017), calling this assumption into question.
The quantitative importance of this selective transfer of INPs from SML to the SSAs is a
topic requiring further research beyond the scope of the current study and is not
accounted for here. Additionally, this approach did not attempt to correct for the possible
entrainment of multiple ice-nucleating entities into a single sea spray particle.
**b.   M18 ice nucleation scheme of MOA**
Another empirical INP parameterization of MOA was derived based on the
correlation between ambient aerosols and INPs measured during the "clean scenario" at
Mace Head Station in August 2015 (McCluskey et al., 2018a, hereafter as M18).
Therefore, M18 includes the effect of physiochemical selective emission and aerosol
chemistry in the air which is missed in W15. This parameterization follows the same
functional form as the surface-active site density ($n_s$) parameterization of Niemand et al.
(2012) for dust, but with different coefficients for MOA, as given below:
$$n_s(T) = e^{(-0.545(T-273.15)+1.0125)} \qquad (8)$$
**c.   N12 ice nucleation scheme of dust**
A surface-active site density-based ice nucleation scheme for immersion freezing
on dust was derived by Niemand et al. (2012) (hereafter referred to as N12) based on
measurements of the AIDA cloud chamber. N12 relates the number concentration of dust
INPs to the dust aerosol number concentration ($N_{tot}$), dust particle surface area ($S_{ae}$,
calculated based on dry diameter of particles), and the density of ice-active surface sites
at a given temperature $T$ ($n_s(T)$), shown as:

$$N_{INP}(T) = N_{tot}S_{ae}n_s(T) \qquad (9)$$
in which  $n_s(T)$  is given as:



$$n_s(T) = e^{(-0.517(T-273.15)+8.934)} \qquad (10)$$

N12 is valid in the temperature range from –36 to –12 °C.

**d. D15 ice nucleation scheme of dust**
As the N12 scheme relates INPs to all sizes of dust aerosol, it may overestimate
INPs, since smaller dust aerosol (<0.5 μm) may not be effective as INPs. An empirical
ice nucleation scheme for the immersion freezing on dust aerosol with sizes larger than
0.5 μm was derived based on field and laboratory measurements (DeMott et al., 2015)
(hereafter referred to as D15). The dust INP number concentration is calculated as

$$N_{INP}(T) = a(n_{0.5})^b e^{c(T-273.15)-d} \qquad (11)$$


where a = 3, b = 1.25, c = –0.46, d =11.6, and $n_{0.5}$ is the number concentration of dust
particles with diameters larger than 0.5 μm.
We note that the above ice nucleation parameterizations (W15, M18, N12, and
D15) are based on empirical formulations. The default heterogeneous ice nucleation
parameterization in CAM6 follows the classical nucleation theory (CNT) (Wang et al.,
2014). CNT is a stochastic scheme that links the freezing rate to the number
concentrations of dust and black carbon aerosols through different heterogeneous ice
nucleation mechanisms (deposition, contact, and immersion). Due to large uncertainties
in heterogeneous nucleation parameterizations, we conducted several ice nucleation
sensitivity experiments in CAM6 as will be discussed in section 2.3.
**2.3 Model configurations and experiments**
In this study, we carried out several numerical experiments to investigate the
influence of MOA on aerosols as well as CCN and INP activities (Table 4). All
simulations were performed for 10 years with prescribed climatological sea surface
temperatures and sea ice. The first year of simulations was treated as model spin-up, and
last nine years of simulations were used in analyses. The simulations were driven by the
present-day (year 2000) aerosol and precursor gas emissions with given greenhouse gas





concentrations. The model was run for 32 vertical levels from surface up to 3 hPa with a
horizontal resolution of 0.9° (latitudes) by 1.25° (longitude). We conducted two sets of
experiments. The first set of experiments, as listed in Table 4, are used to test the model
sensitivity to different MOA emission schemes. The baseline experiment (BASE) uses
the default CAM6 model which does not account for MOA emission and related physical
processes. In addition to the BASE experiment, the B14 experiment addresses emission,
advection, dry/wet deposition, and CCN effect of MOA using the Burrows et al. (2014)
emission scheme. We also designed two additional experiments (G11 and NULL) to
address the model sensitivity to emission methods. These simulations (B14 and G11)
were conducted with the added and internally-mixed MOA approach, following Burrows
et al. (2018). The INP effect of MOA is not considered in this set of experiments.
We conducted another set of experiments to investigate both CCN and INP effects
of MOA, as listed in Table 4. The control experiment (CTL) is the same as BASE except
that the D15 dust ice nucleation scheme was used to replace the CNT scheme in BASE,
because D15 gave a better model performance compared with observations in our
previous study (Shi and Liu, 2019). The B14_D15, which is based on CTL, considers the
MOA emission from B14 and the CCN effect of MOA. The B14_D15_M18 experiment,
which is based on B14_D15, additionally considers the INP effect of MOA based on
M18. The comparison between CTL and B14_D15 shows the CCN effect of MOA, while
the comparison between B14_D15 and B14_D15_M18 shows its INP effect.
We further conducted three experiments to examine the model sensitivity to a
different MOA ice nucleation parameterization (i.e., W15) in B14_D15_W15, and to two
different dust ice nucleation parameterizations (i.e., N12 and CNT) in B14_N12_M18
and B14_CNT_M18 by comparing them with the B14_D15_M18 experiment,
respectively.

## 3 Results

### 3.1 Evaluation of modeled MOA

Given that a realistic representation of MOA emissions is a prerequisite for
models to quantify its influence on ice nucleation, we evaluate three different MOA



388 emission parameterizations in this section. We also analyze the processes contributing to

389 MOA burden such as emission, transport, and removal, because the burden pattern

390 largely determines the INP distribution pattern. Comparisons with available observations

391 are made to examine the performance of different MOA emission schemes.

392  Table 5 lists the annual global mean emissions and burdens of MOA and sea salt

393 from different simulations. Overall, the G11 method generates the largest global MOA

394 emission (27.1 Tg yr$^{-1}$) followed by the B14 method (24.5 Tg yr$^{-1}$). The magnitudes of

395 MOA emissions are within the range of previous studies (Huang et al., 2018; Meskhidze

396 et al., 2011; Langmann et al., 2008). The ratios of MOA emission to sea salt emission are

397 0.67% and 0.74% for the B14 and G11 experiments, respectively, which are also

398 comparable to previous studies ranging from 0.3% to 3.2% (Huang et al., 2018;

399 Meskhidze et al., 2011). The NULL approach only gives an annual global emission of 4.6

400 Tg yr$^{-1}$, with the ratio of MOA emission to sea salt emission of 0.13%. These values are

401 much lower than those of B14 and G11 approaches. We further evaluate aerosol mass

402 mixing ratios and number concentrations in each aerosol mode in the B14 experiment,

403 where MOA is added and internally mixed with sea salt. In B14, MOA comprises up to

404 70% and 50% of Aitken and accumulation mode SSA mass, respectively. Number

405 concentrations of accumulation mode aerosols near the surface are increased by up to 50%

406 over some regions of the Southern Ocean and Arctic.

407  Despite the fact that there are differences in the global annual mean value, B14

408 and G11 generate similar spatial patterns of MOA emission rates (Fig. 1), while G11

409 tends to give higher emission rates than B14. Large emission rates are located in the

410 mid-latitude storm tracks, equatorial upwelling, and coastal regions as shown in Fig. 1.

411 These locations largely reflect the geographic distribution of primary ocean productivity

412 as indicated by [Chl-a] (in G11) or organic matter concentrations (in B14).

413  Here we illustrate the influence of surface wind speeds (supplemental Fig. S1) on

414 the emission of MOA. Although high MOA emissions are mostly co-located with

415 vigorous oceanic biological activities, the oceanic area with smaller/larger wind speed

416 tends to have a decreased/elevated emission rate relative to their biological activities. For

417 instance, due to weak wind speeds (~5 m s$^{-1}$), a strong signal of oceanic organic matter

418 concentration does not correspond to a large emission rate in the west coast of South



America. On the contrary, because of strong wind speeds (~10 m s⁻¹), moderate emission
rates are noticed over the subtropical North Pacific Ocean and subtropical South Indian
Ocean despite relatively small [Chl-a] or organic matter concentrations. This wind speed
dependent pattern is more clearly shown in the B14 results than in the G11 results,
because in the B14 emission scheme, $F_{MOA/SSA}$ is not related to the wind speed while
SSA emission is proportional to the surface wind speed, as described in section 2.2.1.
Conversely, $F_{MOA/SSA}$ is inversely related to the wind speed in G11, results in a more
complicated relationship between wind speed and MOA emission rate in G11.

The global mean MOA burden is 0.097 Tg in B14, which is in close agreement

with previous studies which suggested a range of 0.031 to 0.131 Tg (Huang et al., 2018;
Burrows et al., 2018). The global distribution of MOA column burden shares the similar
patterns between G11 and B14, with the peak burden around 1 mg m⁻² over the mid-to
high latitude Southern Ocean (Fig. 1). Despite the fact that large burdens are usually
related to locations of high emissions, they are also influenced by advection (dependent
on 3-D wind), dry deposition (dependent on particle size), and wet deposition (dependent
on precipitation). The oceanic regions with small annual precipitation rates (supplemental
Fig. S1) lead to considerable accumulations of MOA in G11 and B14. For instance, the
peak burdens with maximum values of 0.4 to 0.6 mg m⁻², on either side of the Pacific
tropical convection zone correspond to the subsidence induced dry zone (i.e., subsiding
branch of Walker and Hadley circulations).

Zonally-averaged vertical distributions of MOA mass mixing ratio illustrate the

vertical transport of MOA (Fig. 1). Simulations from G11 and B14 exhibit a maximum
value of 0.35 μg kg⁻¹ within the boundary layer, located in 40°–50°S of the Southern
Ocean, while the maximum value is only 0.05 μg kg⁻¹ in NULL. Globally, G11 shows
slightly higher MOA mass mixing ratios over all latitudes compared with B14, and
transports more MOA to high altitudes over the tropical regions. It is clear that MOA is
accumulated in the lower troposphere, i.e. below 600 hPa in G11 and B14, and below 800
hPa in NULL. The reason is that MOA is generated over the oceans, especially over the
storm track regions with high precipitation, limiting MOA mainly to the lower
troposphere.



We further evaluate model simulated MOA concentrations with measurements at

Mace Head (Ireland) and Amsterdam Island (Fig. 2). The B14 and G11 methods do well
in capturing the observed seasonal variation of MOA concentrations at Amsterdam Island
(Fig. 2a), although the model produces slightly higher MOA concentrations. At Mace
Head, the two methods produce delayed concentration peaks by about one month
compared with observations (Fig. 2b). The mass fraction of MOA in SSA (Fig. 2c) shows
a better agreement between the model and observation. Both the simulated and observed
organic mass fraction increase from March and reaches a peak in July, although the
observed peak is broader. The NULL approach does not reproduce observed seasonal
cycles of MOA and significantly underestimates observed MOA concentrations due to
the prescribed mass fraction (0.05) in the accumulation mode.

Based on our analyses and comparisons with observations, we show that B14

implementation of MOA emission into CAM6 reasonably captures the concentrations and
seasonal variations of MOA. Next we will study the MOA effects on clouds with a focus
on its INP effect, based on model experiments with the B14 emission (Table 4).

## 3.2 Impact of MOA on CCN

After introducing MOA in the model, we notice an obvious increase in oceanic

surface CCN concentrations at high latitudes. Figure 3 shows the spatial distribution of
annual mean percentage changes in surface CCN concentrations at a supersaturation of
0.1% due to MOA, derived from the two experiments (CTL and B14_D15). From Fig. 3,
the annual mean CCN concentration increases by 15%–35% over much of the oceans
from 30°S to 70°S, with a maximum increase of 45% located over the Southern Ocean
(60°S, 55°E). Other regions showing significant increases of CCN are over the pristine
high latitudes, with increases of 25–35% from 60°S to Antarctica in the SH and from
60°N to 80°N in the NH. These results are comparable with previous results with an
average increase by 12% and up to 20% of CCN over the Southern Ocean (Meskhidze et
al., 2011). Over low- and mid-latitude oceans, CCN changes due to MOA are smaller.
Generally, the distribution of CCN change is consistent with the MOA emission pattern.
The vertical profiles of CCN concentrations from the two model experiments and
observations during the eight field campaigns are shown in Fig. 3. Clear increases of





CCN concentrations in the boundary layer due to MOA are evident for campaigns over
the ocean or coastal regions (SOCEX1, SOCEX2, ACE1, FIRE1, and ASTEX), with a
maximum increase (26%) in ACE1. Observed CCN from FIRE1 shows a strong
inversion of CCN below 800 hPa, and this inversion is challenging for the model due to
its coarse vertical resolution. An obvious underestimation of CCN in the model is noticed
at FIRE3 over the Arctic Ocean in Spring, which is attributed to the underestimated
transport of air pollution caused by too strong wet scavenging in the model (Liu et al.,

2012).

## 3.3 Impact of MOA on INPs

In order to examine the importance of MOA INPs, we compare modeled INPs

from MOA versus dust as well as compare them with observations from several field
campaigns in high latitudes (Fig. 4). Modeled INP concentrations from MOA are
calculated online using M18 and W15 parameterizations (from B14_D15_M18 and
B14_D15_W15 experiments, respectively), while dust INP concentrations are calculated
online using D15, CNT, and N12 parameterizations (from B14_D15_M18,
B14_CNT_M18, and B14_N12_M18 experiments, respectively). Modeled INP
concentrations are computed based on aerosol concentrations at different temperatures
and are selected at the same altitudes and locations as the observations. The measured
INP data were obtained from Mace Head, the CAPRICORN campaign (Clouds, Aerosols,
Precipitation, Radiation, and Atmospheric Composition over the Southern Ocean),
Oliktok Point, Zeppelin, and the SOCRATES campaign (Southern Ocean Clouds,
Radiation, Aerosol Transport Experimental Study) (McCluskey et al., 2018a; McCluskey
et al., 2018b; Creamean et al., 2018; Tobo et al., 2019).

As illustrated in Fig. 4, the M18 parameterization tends to underestimate observed

INP concentrations except at temperatures colder than –25°C. On the other hand, the W15
parameterization overestimates observed INP concentrations except at temperatures
warmer than –20°C. Under the same MOA scenario, the W15 parameterization is more
efficient in producing INPs than M18. This is because the M18 parameterization was
derived from MOA in the atmosphere which accounts for the effect of physiochemical
selective emission and aerosol chemistry in the air. In contrast, the W15 parameterization



was derived based on the total organic carbon in sea surface microlayer samples, which
contain higher organic mass concentrations compared with ambient MOA.

The dust INP concentration calculated with CNT shows an underestimation when

temperature is warmer than –20℃ and an overestimation when temperature is between
–30℃ and –20℃. This is consistent with previous work by Wang et al. (2014). The D15
parameterization indicates a clear underestimation. Meanwhile, the N12 parameterization
reveals an overall overprediction of INPs compared with observations. These results
suggest that the N12 parameterization is more efficient in producing dust INPs than the
D15 parameterization under the same dust loading. INP concentrations from N12 are
calculated based on the coarse, accumulation, and Aitken mode dust aerosol, which
account for fine dust particles, while INP concentrations from D15 are calculated based
on the number concentration of dust particles with diameters larger than 0.5 μm (DeMott
et al., 2015). Simulated total INPs, the sum of dust and MOA INPs from D15 and M18,
gives a better agreement with observations than D15 and M18 alone, although
underestimations still exist at warmer temperatures.

Fig. 5 shows the comparison between simulated and measured INPs from five

parameterization schemes as a function of temperature for the same field campaigns as in
Fig. 4. Generally, an inverse linear relationship is revealed between $\log_{10}$(INPs) and
temperature in the measurements. This relationship is also shown in simulated INP number
concentrations from the empirical parameterizations (N12, D15, W15, M18). However, for
CNT, nearly constant INP number concentrations are presented at temperatures from
–35℃ to –20℃, and then a rapid decrease with increasing temperature when temperature
is warmer than –20℃. At temperatures higher than –15℃, nearly no INPs are produced by
CNT, leading to the underestimation of INPs in the CNT method at these temperatures.

We notice higher INP number concentrations are produced from M18 compared

with W15 at Zeppelin during March 2017. The most distinctive feature of this campaign is
its very low aerosol loadings. For example, simulated SSA mass mixing ratio is around 0.6
μg kg⁻¹ with the maximum value at 1.8 μg kg⁻¹ below 850 hPa, and the dust mass mixing
ratio is around 0.3 μg kg⁻¹. We note that simulated dust INP number concentrations from
N12 are always higher than those from D15, and both N12 and D15 are more efficient in
producing INPs than CNT when temperature is warmer than –20℃.


The global distribution pattern of annual mean MOA INP concentrations at 950
hPa at temperature of –25℃ is similar to that of MOA column burden concentrations, as
shown in Fig. 6a. The MOA INPs are spread over the oceans, with peaks (~0.1 L⁻¹) over
40°S to 60°S of the Southern Ocean, the subtropical Southern Indian Ocean, the
subtropical Atlantic Ocean, and the subtropical Eastern Pacific Ocean. Meanwhile, dust
INP concentrations diagnosed at the same pressure and at the same temperature (Fig. 6b)
are dominant over the NH and downwind of dust source regions in the SH (e.g., around
Australia and extended to 50°S).
Fig. 6c shows the horizontal distribution of ratio of MOA INP concentration to
dust INP concentration at 950 hPa. It is clear that MOA INPs are more important than
dust INPs in the 40°S south of SH, where MOA INP concentrations can reach up to 1000
times higher than those of dust INPs. The zonal mean vertical distribution of ratio of
MOA INP concentration to dust INP concentration is illustrated in Fig. 6d. The ratio
peaks near 65°S, indicating that MOA INPs are more important than dust INPs over the
Southern Ocean from surface up to 400 hPa, and extends poleward to 90°S. Above the
400 hPa altitude, dust particles are still more important INPs. Because dust particles are
emitted over drier deserts (i.e., with lower precipitation and thus less wet scavenging),
dust can be subject to long-range transport at high elevations. In contrast, most MOA
particles are generated over the storm track regions with high occurrences of precipitation.
Taking into account of emission, transport and wet scavenging of MOA and dust particles
results in MOA INPs dominating below 400 hPa over the Southern Ocean while dust
INPs are generally more important elsewhere.
Immersion freezing on MOA in mixed-phase clouds requires that there are cloud
droplets at temperatures colder than –4℃. Ice nucleation consumes cloud liquid water, and
thus will compete with other processes for cloud liquid water (e.g., autoconversion of
cloud water to rain, accretion of cloud water by rain and snow). This competition is
expected to result in a reduction of ice nucleation rate of MOA compared with the offline
calculation of ice nucleation rate as in McCluskey et al. (2019). Fig. 7 shows the annual
zonal mean ice production rates from the immersion freezing of MOA and dust, which are
calculated online for the cloud ice production tendency in the B14_D15_M18 experiment.
Over the NH, the immersion freezing of dust dominates the primary ice production, giving





an averaged ice production rate at 5 kg$^{-1}$s$^{-1}$ and up to 20 kg$^{-1}$s$^{-1}$ over 40°N at 400 hPa (Fig.
7b), while the MOA ice production rate is around 1 kg$^{-1}$s$^{-1}$ (Fig. 7a). However, in the
Arctic boundary layer, the MOA fraction of total ice production rate is around 0.6~0.7 (Fig.
7c), indicating that MOA INPs are more important in generating ice crystals than dust INPs
there. Over the SH, the immersion freezing rate of MOA dominates the primary ice
production below 400 hPa with the MOA fraction close to 1. The zonal average ice
nucleation rate of MOA is around 1 kg$^{-1}$s$^{-1}$, and up to 5 kg$^{-1}$s$^{-1}$ over the 65°S Southern
Ocean at 400–600 hPa. The immersion freezing rate of dust is around 1 kg$^{-1}$s$^{-1}$ above 500
hPa, and smaller than 0.1 kg$^{-1}$s$^{-1}$ below 600 hPa altitude in the SH. Analysis of the seasonal
variation of ice nucleation rate of MOA indicates that a maximum rate of about 16 kg$^{-1}$s$^{-1}$
occurs at 400–600 hPa over 60°S in July (austral winter). In summary, the annual mean
immersion freezing of MOA dominates the primary ice production over the SH below 400
hPa altitude and in the Arctic boundary layer.

### 584 3.4 Impact of MOA on clouds and radiative forcing

Table 6 displays the differences of cloud and precipitation variables between the
CTL and B14_D15_M18 experiments. With added MOA aerosol, the global annual mean
surface concentration of CCN at 0.1% supersaturation changes from 103.3 cm$^{-3}$ in CTL to
106.6 cm$^{-3}$ in B14_D15_M18. This increase of 3.28 cm$^{-3}$ is comparable to other model
estimates of 3.66 cm$^{-3}$ (Burrows et al., 2018), and 2.6–3.0 cm$^{-3}$ (Meskhidze et al., 2011).
The vertically-integrated cloud droplet number concentration (CDNUMC) increases by
5.25% in B14_D15_M18 compared with CTL, and by up to 16.89% over 20–90°S during
the austral summer (December-January-February). The global annual mean liquid water
path (LWP), ice water path (IWP), longwave cloud forcing (LWCF), and total cloud
fraction (CLDTOT) do not show obvious changes between CTL and B14_D15_M18. The
global annual mean shortwave cloud forcing is stronger by –0.41 W m$^{-2}$ due to MOA.
During the austral summer over 20–90°S, we notice an increase of 4.57 g m$^{-2}$ (5.10%) in
LWP, and a 1.35% (2.52%) increase in low-cloud fraction. As a consequence, SWCF is
enhanced by –2.87 W m$^{-2}$ (Table 6), which is comparable to –3.5 W m$^{-2}$ estimated in
Burrows et al. (2018). Ice number concentration on –15°C isotherm increases by 9.34%





during the austral winter. There does not appear to be a significant change in LWCF, which
is consistent with the result of Huang et al. (2018).
Strong CCN effect of MOA on clouds (in terms of significant changes in CCN and
CDNUMC) tends to occur only in the SH over 40–60°S, while strong INP effect (in terms
of significant changes in cloud ice mass and number concentrations) is notable over 50–70°
in both Hemispheres (Fig. 8). Over 40–60°S, a significant increase from 70 to 90 cm$^{-3}$ in
the annual zonal mean surface CCN concentration is observed. The CCN concentration
there is nearly 30% higher in B14_D15 and B14_D15_M18 than in CTL. As a result,
CDNUMC increases from $2.6 \times 10^{10}$ m$^{-2}$ in CTL to $3.0 \times 10^{10}$ m$^{-2}$ in B14_D15 and
B14_D15_M18 over 40–60°S, leading to an increase in LWP due to the aerosol indirect
effect (Fig. 8). Furthermore, we notice a stronger SWCF at 40–60°S by 3 W m$^{-2}$ in
B14_D15 compared with CTL. After considering the INP effect of MOA in the model, we
notice that cloud ice number concentration and cloud ice mass mixing ratio increase in
mixed-phase clouds which led to a slightly decrease in CDNUMC. As indicated in Fig.
8b,d, cloud ice number concentration increases from 4500 kg$^{-1}$ in B14_D15 to 5500 kg$^{-1}$ in
B14_D15_M18 at ~60°S, with cloud ice mass mixing ratio increased by 0.25 mg kg$^{-1}$.
Over 60°N, cloud ice number concentration increases from 4200 kg$^{-1}$ in B14_D15 to 5200
kg$^{-1}$ in B14_D15_M18, with cloud ice mass mixing ratio increased by 0.1 mg kg$^{-1}$.
Fig. 9 shows the seasonal variations of cloud properties and cloud radiative forcing
averaged over the 20°S–90°S in SH, in response to the introduction of MOA as CCN and
INPs. The seasonal variation of surface CCN concentration at 0.1% supersaturation shows
the maximum value of 72 cm$^{-3}$ in the austral summer and the minimum value of ~50 cm$^{-3}$
in the austral winter in CTL. Similar seasonal variation patterns are also noted for
CDNUMC and LWP. With the inclusion of MOA in the model, B14_D15 and
B14_D15_M18 produce more surface CCN, with an increase of up to 14 cm$^{-3}$ (~20%) in
January, compared with CTL. Accordingly, CDNUMC increases from $2.1 \times 10^{10}$ m$^{-2}$ in
CTL to $2.5 \times 10^{10}$ m$^{-2}$ in B14_D15 in January, and LWP increases from 93 g m$^{-2}$ in CTL to
97 g m$^{-2}$ in B14_D15 in January. As a consequence, SWCF is stronger by –3.5 W m$^{-2}$ in
B14_D15 compared with CTL during the austral summer. We also notice that CCN,
CDNUMC, and SWCF show smaller changes during the austral winter due to weaker
oceanic biological activity.





631  Different from the warm cloud features above, seasonal variations of ice properties in

632 mixed-phase clouds (i.e., cloud ice mass mixing ratio and number concentration on –15°C

633 isotherm, IWP) clearly show winter maxima. After introducing the INP effect of MOA in

634 the model, ice number concentration on –15°C isotherm increases by comparing B14_D15

635 with B14_D15_M18, with obvious increases of up to 27% in June. Ice mass mixing ratio

636 on –15°C isotherm increases by 0.19 mg kg$^{-1}$ (13%) in June. Increases in both cloud ice

637 number and mass contribute to the increase of IWP by 0.5 g m$^{-2}$ in austral winter. The

638 seasonal change of LWCF is not well correlated with changes in ice number concentration

639 and mass mixing ratio in mixed-phase clouds, because LWCF is controlled more by high

640 clouds. Our introduction of MOA INPs mainly occurs in mixed–phased clouds, and

641 therefore has a small influence on LWCF.

642  As shown in Table 7, the CCN effect of MOA on SWCF is strongest in the austral

643 summer, with the value of –2.78 W m$^{-2}$ over the 20°S–90°S in SH. In contrast, the INP

644 effect of MOA on LWCF is strongest in the austral winter, with the value of 0.35 W m$^{-2}$

645 (Table 8). For the net cloud forcing (SWCF + LWCF), the CCN effect of MOA is 2.65 W

646 m$^{-2}$ in the austral summer, and the INP effect is 0.65 W m$^{-2}$ in austral spring over the

647 20°S–90°S. The annual global mean CCN and INP effects of MOA on the net cloud

648 forcing are –0.35 and 0.016 W m$^{-2}$, respectively. From an annual mean perspective, the

649 CCN effect of MOA on SWCF is –0.84 W m$^{-2}$ over 20–90°S and is about twice as much as

650 the global mean value (–0.41 W m$^{-2}$), which indicates that the global annual mean SWCF

651 change due to MOA is dominated by SH contributions.

652 **4 Discussion**

653  In this study, for the MOA emission process, we only considered the generation

654 of MOA during the film drop breakup in B14, and the generation of MOA from jet drops

655 is not currently included. The film drops form from bubble-cap films bursting, while the

656 jet drops generate from the base of breaking bubbles.  Particles from jet drops, with the

657 diameter is around supermicrometer, are considered larger than particles from film drops

658 (Wang et al., 2017). Extending the current emission scheme to include MOA emissions

659 through jet drops (Wang et al., 2017) may be possible with more measurements and an



improved understanding of physical mechanisms that determine the sea spray organic
emission.
For the ice nucleation efficiency of MOA, the M18 parameterization only
includes the more persistent, heat-stable component observed in ambient sea spray
aerosol INP sampling. This neglects the heat-labile organic INPs (McCluskey et al.,
2018b). Regarding ice nucleation mechanisms, only the immersion mode of ice
nucleation is implemented in this study, however, recent laboratory experiments (Wolf et
al., 2019) have indicated a potentially important role of MOA in the deposition mode at
temperatures below –40°C. Future work will focus on improving the limitations of the
current understanding of MOA emission and ice nucleation in the model.
Recent studies indicated an underestimation of ice formation in CAM6
(D'Alessandro et al., 2019) that results in too much cloud liquid and too little cloud ice in
mixed-phase clouds. In addition to ice nucleation undertaken in this study, other factors
may contribute to this model bias. For example, the CLUBB scheme used in CAM6 for
turbulence and shallow convection treats only liquid phase condensation, lacking ice
formation in the model's large-scale cloud macrophysics (Zhang et al., 2020).
Furthermore, CAM6 misses the representation of several important mechanisms of
secondary ice formation. Observed secondary ice formation processes include rime
splintering, ice-ice collision fragmentation, droplet shattering during freezing, and
fragmentation during sublimation of ice bridges (Field et al., 2017). Currently, only the
rime splintering is considered in CAM6. Lastly, CAM6 with a horizontal resolution of
approximately 100 km may not resolve the subgrid cloud processes and heterogeneous
distributions of cloud hydrometeors (Tan et al., 2016; Zhang et al., 2019). These issues
will be addressed in future studies.
**5 Summary and Conclusions**
This study introduces MOA into CAM6 as a new aerosol species and treats the
chemistry, advection, and wet/dry deposition of MOA in the model. This paper also
considers the MOA influences on droplet activation and ice nucleation, particularly
focusing on quantifying the INP effect of MOA on cloud properties and radiation. Here
we summarize our main findings:


(1) Three different emission schemes (B14, G11, and NULL) of MOA were
implemented in the model and simulated MOA concentrations were evaluated with
available observations. The global simulation indicates that high MOA burden centers are
mostly co-located with regions of vigorous oceanic biological activities and high wind
speeds such as in mid-latitude storm tracks, the equatorial upwelling, and coastal regions.
The global MOA emission is 24.5 Tg yr$^{-1}$ in B14, 27.1 Tg yr$^{-1}$ in G11, and 4.6 Tg yr$^{-1}$ in
the NULL emission approach. On the global scale, the MOA mass emission is 0.67%,
0.74%, and 0.13% of the sea salt mass emission from B14, G11, and NULL, respectively.
We show that observed seasonal cycles of marine organic matter at Mace Head and
Amsterdam Island are reproduced when the MOA fraction of SSA is assumed to depend
on sea spray biology (B14, G11), but are not reproduced when this fraction is assumed to
be constant (NULL). Our study does not support the constant organic mass fraction of
SSA emissions (Quinn et al., 2014; Saliba et al., 2019; Bates et al., 2020).
(2) After introducing MOA in the model, annual mean CCN concentrations (at
supersaturation of 0.1%) are increased by 15%–30% over the oceans ranging from 30°S
to 70°S. Two different ice nucleation schemes of MOA (M18 and W15) are implemented
and compared with available measurements. The INPs from MOA by the M18
parameterization show a reasonable agreement with observations at NH and SH high
latitudes, while simulated total INPs, the sum of MOA INPs from M18 and dust INPs
from D15, give a better agreement with observations. W15 for MOA alone overestimates
the observed INP concentrations across all temperatures. At –25℃, MOA INP
concentrations can be 1000 times higher than those of dust INPs over 40–60ºS in the SH
boundary layer while dust INP concentrations are higher above 400 hPa altitude over SH
and NH.
(3) We notice a strong CCN effect of MOA over 40–60ºS only in SH, while a
strong INP effect of MOA is identified over 50–70º in both Hemispheres. For seasonal
variations, CCN effect is stronger during the austral summer than winter, while INP
effect is stronger in the austral winter than summer. The CCN effect of MOA on SWCF
is strongest in the austral summer over SH with a value of –2.78 W m$^{-2}$, while the INP
effect on LWCF is strongest in the austral winter over SH with a value of 0.35 W m$^{-2}$.
The annual global mean CCN and INP effect of MOA on the net cloud forcing is –0.35 and





0.016 W m$^{-2}$, respectively. This work is a stepping stone towards better climate models
because the important role of MOA in biogeochemistry, hydrological cycle, and climate
change.


**Competing interests:** The authors declare that they have no conflict of interest.

**Data availability:** The model code is available at
https://github.com/CESM-Development. The observed INP data is available at
https://data.eol.ucar.edu/master_lists/generated/socrates/.

**Author contributions**: XZ and XL conceptualized the analysis and wrote the manuscript
with input from the co-authors. XZ modified the code, carried out the simulations, and
performed the analysis. SB provided scientific suggestions to the manuscript and
provided the model code for the emission of marine organic aerosol. YS provided help in
setting up the global climate model, designing the model runs, and created Figures. XL
was involved with obtaining the project grant, supervised the study. All authors were
involved in helpful discussions and contributed to the manuscript.

**Acknowledgment:** This research was supported by the DOE Atmospheric System
Research (ASR) Program (grant DE-SC0020510). S. M. Burrows was also funded by the
U.S. DOE Early Career Research Program. We thank Christina McCluskey for providing
the INP data from the SOCRATES campaign.





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



Table 1. Aerosol species in MAM4 modes

|  | **Accumulation** | **Aitken** | **Coarse** | **primary** |
|---|---|---|---|---|
| Species | num_a1, so4_a1, pom_a1, soa_a1, bc_a1, dst_a1, ncl_a1, mom_a1 | num_a2, so4_a2, soa_a2, ncl_a2, dst_a2, mom_a2 | num_a3, dst_a3, ncl_a3, so4_a3 | num_a4, pom_a4, bc_a4, (mom_a4 if internal added) |
| Size range | $0.08 - 1$ μm | $0.02 - 0.08$ μm | $1-10$ μm | $0.08 - 1$ μm |
| Standard Deviation $\sigma g$ | 1.6 | 1.6 | 1.2 | 1.6 |
| Number-median diameter $Dgn$ | $1.1 \times 10^{-7}$ | $2.6 \times 10^{-8}$ | $2.0 \times 10^{-6}$ | $5.0 \times 10^{-8}$ |
| Low bound $Dgn$ | $5.35 \times 10^{-8}$ | $8.7 \times 10^{-9}$ | $1.0 \times 10^{-6}$ | $1.0 \times 10^{-8}$ |
| High bound $Dgn$ | $4.4 \times 10^{-7}$ | $5.2 \times 10^{-8}$ | $4.0 \times 10^{-6}$ | $1.0 \times 10^{-7}$ |



Table 2. Aerosol species and physical properties

| Species | Name | Density (kg m$^{-3}$) | Hygroscopicity |
|---------|------|----------------------|----------------|
| BC | Black carbon | 1700 | $1.0 \times 10^{-10}$ |
| SO4 | Sulfate | 1770 | 0.507 |
| SOA | Secondary organic | 1000 | 0.14 |
| POA | Primary organic | 1000 | $1.0 \times 10^{-10}$ |
| DST | Dust | 2600 | 0.068 |
| NCL | Sea salt | 1900 | 1.16 |
| MOA | Marine organic aerosol | 1601 | 0.1 |



Table 3. Molecular weights, mass at saturation, Langmuir parameters of the three ocean macromolecules

| Species | polysaccharides | proteins | Lipids |
|---|---|---|---|
| Molecular weight $[g\ mol^{-1}]$ | 250000 | 66463 | 284 |
| mass per area at saturation $[g\ m^{-2}]$ | 0.1376 | 0.00219 | 0.002593 |
| Langmuir parameter $[m^3\ mol^{-1}]$ | 90.58 | 25175 | 18205 |





Table 4. List of experiments to test model sensitivity to different emission and ice nucleation schemes

| Name | Emission of MOA | DUST ice nucleation | MOA ice nucleation | Notes |
|---|---|---|---|---|
| BASE | — | CNT | — | Base line simulation |
| B14 | Burrows et al. [2014] | CNT | — | Sensitivity test of emission scheme |
| G11 | Gantt et al. [2011] | CNT | — | Sensitivity test of emission scheme |
| NULL | NULL | CNT | — | Sensitivity test of emission scheme |
| CTL | | DeMott et al. [2015] | | Control simulation |
| B14_D15 | Burrows et al. [2014] | DeMott et al. [2015] | | CCN effect |
| B14_D15_M18 | Burrows et al. [2014] | DeMott et al. [2015] | McCluskey et al. [2018] | INP effect |
| B14_D15_W15 | Burrows et al. [2014] | DeMott et al. [2015] | Wilson et al. [2015] | Sensitivity test of MOA INP parameterization |
| B14_N12_M18 | Burrows et al. [2014] | Niemand et al. [2012] | McCluskey et al. [2018] | Sensitivity test of dust INP parameterization |
| B14_CNT_M18 | Burrows et al. [2014] | CNT | McCluskey et al. [2018] | Sensitivity test of dust INP parameterization |



Table 5. Annual global mean emissions and burdens of MOA and sea salt

| Name | Sea salt emission ($Tg\ yr^{-1}$) | MOA emission ($Tg\ yr^{-1}$) | Sea salt burden ($Tg$) | MOA burden ($Tg$) | MOA/Sea salt emission (%) |
|---|---|---|---|---|---|
| BASE | 3651 | — | 8.83 | — | — |
| B14 | 3656 | 24.5 | 8.88 | 0.097 | 0.67 |
| G11 | 3666 | 27.1 | 8.86 | 0.120 | 0.74 |
| NULL | 3648 | 4.6 | 8.85 | 0.018 | 0.13 |




Table 6. Mean changes and relative changes (%) between CTL and B14_D15_M18 experiments. Included in the table are surface CCN concentrations at 0.1% (CCN), ice particle number concentration at $-15^\circ$C thermal level (Ni_15), vertically-integrated cloud droplet number concentration (CDNUMC), total grid-box cloud liquid water path (LWP), total grid-box cloud ice water path (IWP), shortwave and longwave cloud forcings (SWCF, LWCF), total cloud fraction (CLDTOT), high/mid-level/low-level clouds (CLDHGH, CLDMED, CLDLOW), and total surface precipitation rate (PRECT), with bold fond indicating relative changes larger than 3%.

|  | Global ANN | 20S–90S ANN | 20S–90S JJA | 20S–90S DJF |
|---|---|---|---|---|
| CCN (cm$^{-3}$) | **3.28 (3.17)** | **4.85 (8.45)** | 1.37 (2.84) | **9.26 (13.47)** |
| Ni_15 (m$^{-3}$) | 39.39 (2.25) | **102.0 (5.21)** | **275.93 (9.34)** | −3.05 (−0.510) |
| CDNUMC (cm$^{-2}$) | **$7.53 \times 10^4$ (5.25)** | **$1.27 \times 10^5$ (8.65)** | $1.10 \times 10^4$ (0.94) | **$3.22 \times 10^5$ (16.89)** |
| LWP (g m$^{-2}$) | 0.69 (1.02) | 0.66 (0.77) | −1.86 (−2.32) | **4.57 (5.10)** |
| IWP (g m$^{-2}$) | 0.05 (0.37) | 0.10 (0.99) | 0.42 (3.69) | 0.13 (1.48) |
| SWCF (W m$^{-2}$) | −0.41 (0.86) | −0.63 (1.17) | 0.400 (−1.48) | **−2.87 (3.47)** |
| LWCF (W m$^{-2}$) | 0.08 (0.35) | 0.031 (0.15) | 0.13 (0.57) | 0.11 (0.52) |
| CLDTOT (%) | 0.12 (0.17) | 0.17 (0.22) | 0.011 (0.014) | 1.05 (1.45) |
| CLDHGH (%) | 0.016 (0.039) | −0.0082 (−0.021) | −0.027 (−0.071) | −0.18 (−0.47) |
| CLDMED (%) | 0.078 (0.26) | 0.19 (0.55) | 0.20 (0.54) | 0.017 (0.054) |
| CLDLOW (%) | 0.13 (0.33) | 0.14 (0.24) | −0.43 (−0.69) | 1.35 (2.52) |
| PRECT (mm day$^{-1}$) | −0.0011 (−0.038) | 0.0042 (0.17) | 0.019 (0.71) | 0.040 (1.66) |



Table 7. CCN and INP effects of MOA on SWCF, and the values in the table are the mean change and
relative change (%). The CCN effect is calculated between CTL and B14_D15 experiments, and the INP
effect is calculated between B14_D15 and B14D15_M18 experiments, with the bold font indicated the
maximum change.

| | | ANN | MAM | JJA | SON | DJF |
|---|---|---|---|---|---|---|
| 20–90S | CCN | −0.84 (1.58) | −0.47 (1.16) | 0.48 (−1.78) | −0.59 (0.95) | **−2.78 (3.36)** |
| | INP | 0.22 (−0.50) | 0.084 (−0.20) | −0.080 (0.30) | 0.94 (−1.51) | −0.088 (0.10) |
| global | CCN | −0.41 (0.85) | −0.21 (0.48) | −0.43 (0.89) | 0.027 (−0.056) | −1.01 (1.96) |
| | INP | −0.0037 (0.0077) | 0.047 (−0.11) | 0.27 (−0.54) | −0.16 (0.33) | −0.17 (0.33) |



Table 8. CCN and INP effect of MOA on LWCF, and the values in the table are the mean change and
relative change (%). The CCN effect is calculated between CTL and B14_D15 experiments, and the INP
effect is calculated between B14_D15 and B14D15_M18 experiments, with the bold fond indicated the
maximum change.

|  |  | ANN | MAM | JJA | SON | DJF |
|---|---|---|---|---|---|---|
| 20–90S | CCN | 0.064 (0.30) | 0.033 (0.15) | −0.21 (−0.93) | 0.29 (1.39) | 0.15 (0.73) |
|  | INP | −0.033 (−0.15) | −0.15 (−0.68) | **0.35 (1.5)** | −0.29 (−1.35) | −0.042 (−0.20) |
| global | CCN | 0.064 (0.27) | −0.0097 (−0.040) | −0.032 (−0.13) | 0.0890 (0.38) | 0.21 (0.91) |
|  | INP | 0.020 (0.085) | −0.12 (−0.50) | 0.21 (0.85) | 0.035 (0.15) | −0.039 (−0.17) |


**Figures**

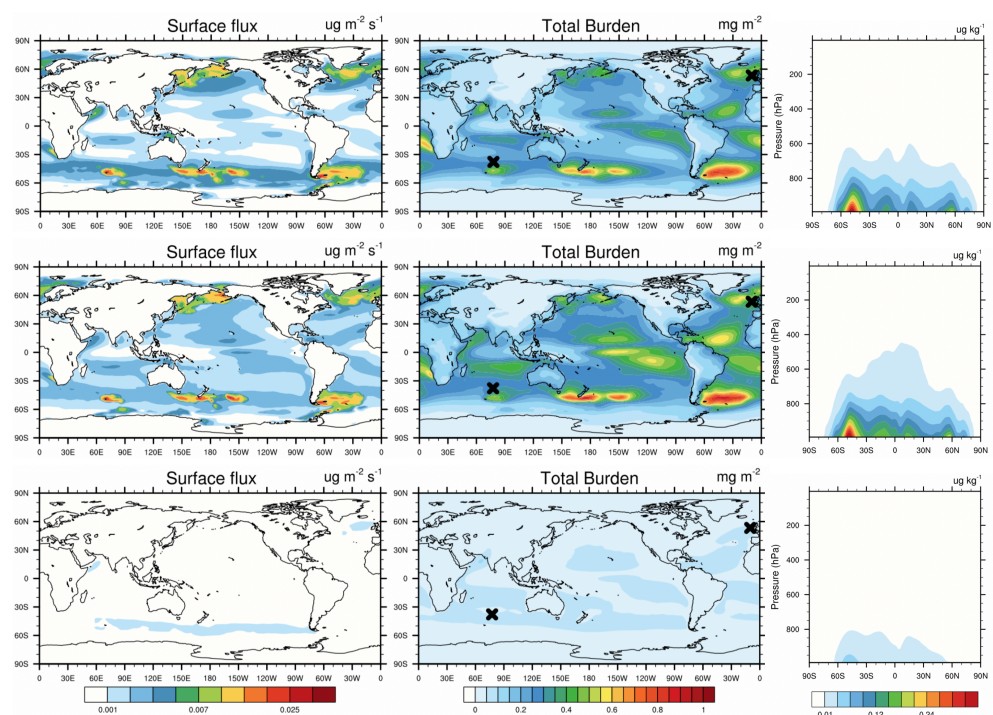

Figure 1. Spatial distributions of annual mean surface flux (first column, in unit of μg m$^{-2}$ s$^{-1}$) and
vertically-integrated (column) burden of MOA (second column, in unit of mg m$^{-2}$), and latitude-pressure
cross-sections of annual mean MOA mixing ratio (third column, in unit of μg kg$^{-1}$) from the B14 (first row),
G11 (second row), and NULL (third row) experiments. The right black cross in the second row indicates
the position of Mace Head, and the left black cross indicates the position of Amsterdam Island.





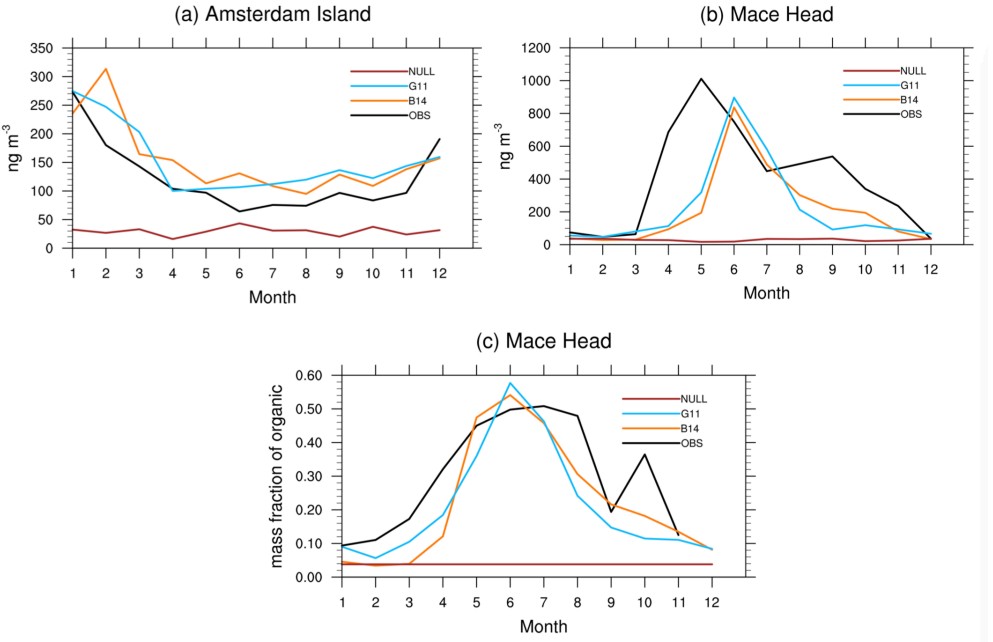

41 Figure 2. Monthly averaged concentrations of MOA at (a) Amsterdam Island and (b) Mace Head Ireland;
42 and (c) monthly averaged mass fraction of MOA in SSA at Mace Head Ireland. The locations of
43 Amsterdam Island and Mace Head Ireland are shown in Figure 1.



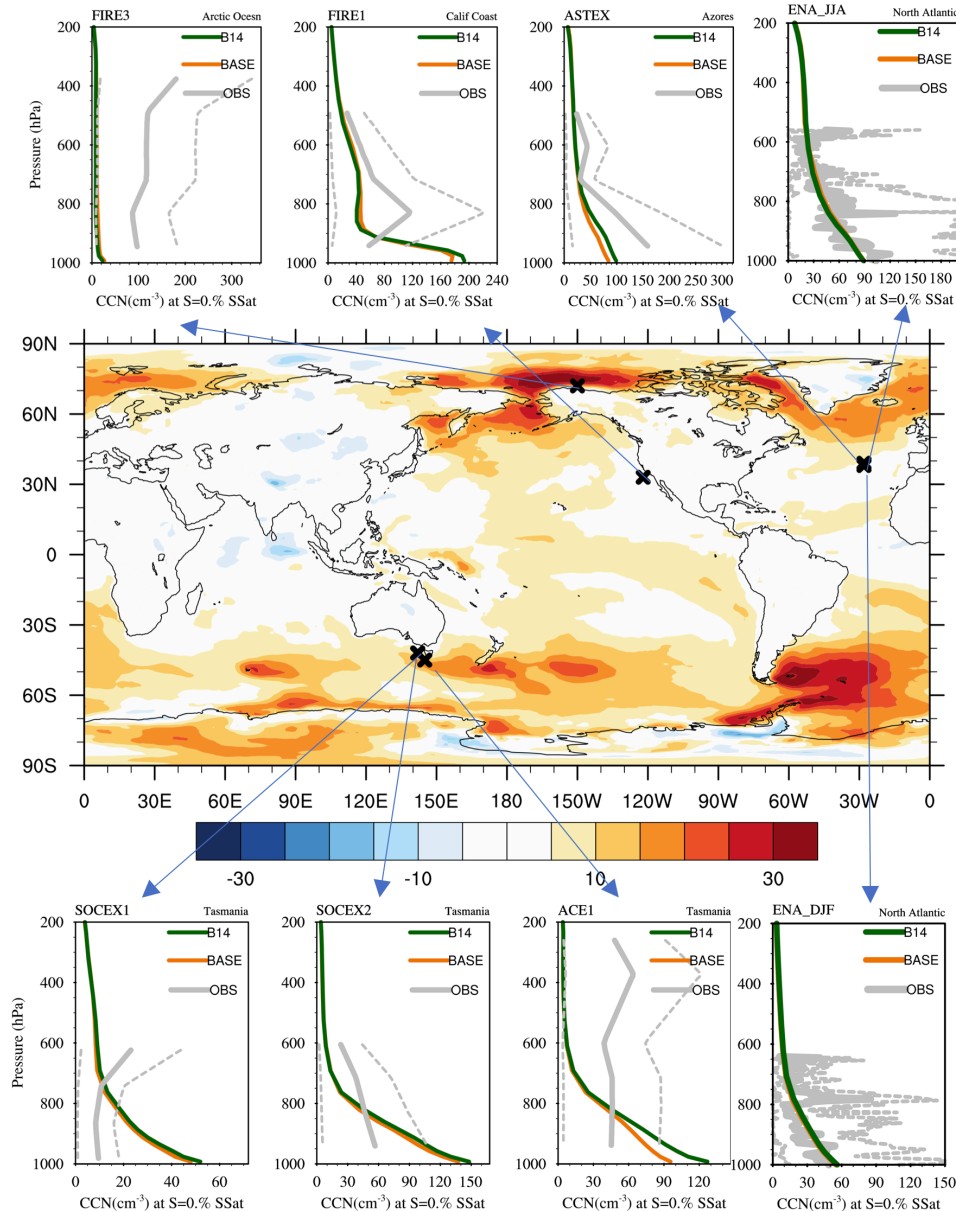

Figure 3. Spatial distribution of annual mean percentage changes of surface CCN concentrations at 0.1%
supersaturation due to MOA, and vertical distribution of CCN concentrations at 0.1% supersaturation from
eight measurements (solid gray lines), BASE (solid orange line) and B14_D15 (solid green line). Dashed
lines outline a range of 10th and 90th percentiles for measurements in different field campaigns: FIRE1
(the First International Satellite Cloud Climatology Project Reginal Experiment) locates at 33° N and
238° W in California coast, the data is collected during June to July, 1987; the FIRE3 locates at 72° N and
210° W in Arctic Ocean, the data is collected during May, 1998; the ASTEX (Atlantic Stratocumulus





Transition Experiment) locates at 38° N and 332° W in Azores, the data is collected during June, 1992; the
SOCEX1 (Southern Ocean Cloud Experiment) is located as –42 ° S and 142° E in Tasmania, the data is
collected during July 1993; the data of SOCEX2 is collected during January to February 1995; the ACE1
(Aerosol Characterization Experiment) locates at –45 ° S , 145° E in Tasmania, the data is collected during
November to December, 1995; and the ENA_JJA(Eastern North Atlantic) locates at 39° N and 332° W in
Eastern North Atlantic, the data is collected during June to August, while ENA_DJF is collected during
December, January, and February, 2006 to 2020.



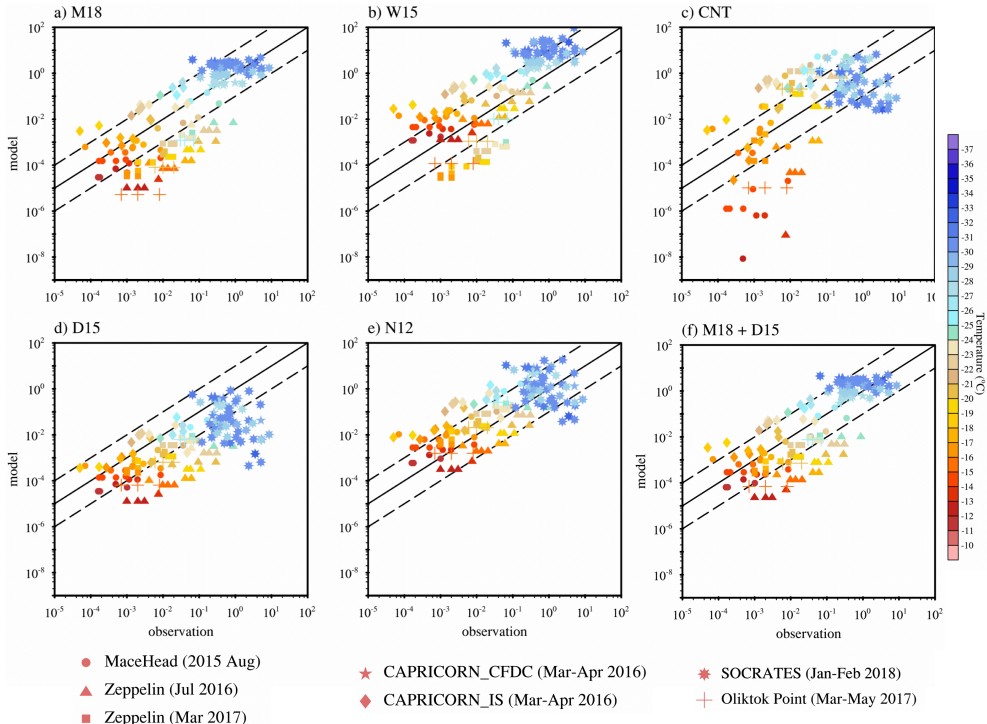

Figure 4. Comparison of simulated vs. observed INP number concentrations for different simulations:
(a) MOA INPs from M18 [McCluskey et al., 2018], (b) MOA INPs from W15 [Wilson et al., 2015],
(c) dust INPs from CNT [Wang et al., 2014], (d) dust INPs from D15 [DeMott et al., 2015], (e) dust
INPs from N12 [Niemand et al., 2012], and (f) sum of dust and MOA INPs from D15 and M18.
Dashed lines outline a factor of 10 about the 1:1 line (solid) in all the panels. Color bar shows the
observed temperature in °C, while different markers represent different field campaigns. Zeppelin site
locates at 78.9081° N, 11.8814° E, 475 m above mean sea level in NyÅlesund, Svalbard, the INP data is
collected during July 2016 and March 2017 [Tobo et al., 2019]; Oliktok Point site locates at 70.50° N
149.89°W, the INP data is collected during March-May 2017 [Creamean et al., 2018)]; CAPRICORN
(Clouds, Aerosols, Precipitation, Radiation, and Atmospheric Composition over the Southern Ocean) INP
data is collected on ships during 13 March to 15 April in 2016 over the Southern Ocean [McCluskey, Hill,
Humphries, et al., 2018a]; Mace Head site locates at 53.32°N, 9.90°W, the INP data is collected during
August 2015 [McCluskey, Ovadnevaite, Rinaldi, et al., 2018b]; SOCRATES (Southern Ocean Clouds,
Radiation, Aerosol Transport Experimental Study) INP data is collected on flights during January-February
2018 over the Southern Ocean by Paul DeMott (https://data.eol.ucar.edu/master_lists/generated/socrates/).



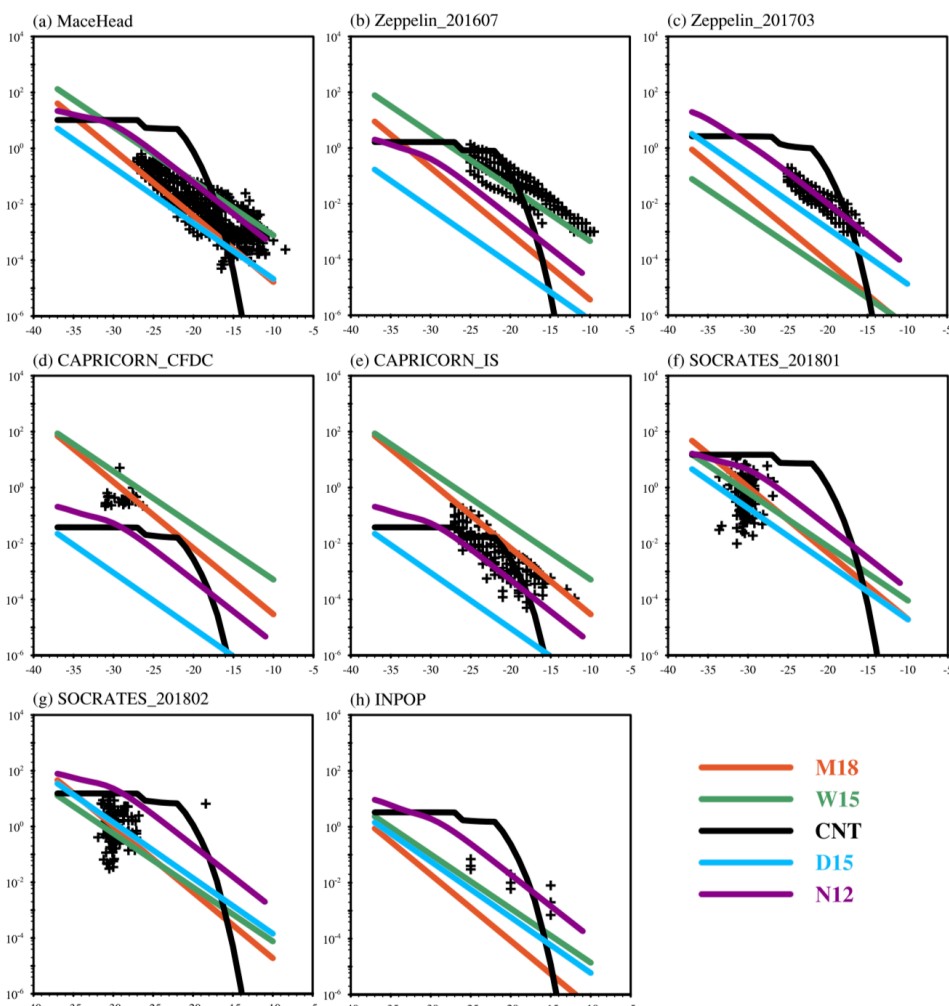

Figure 5. Modeled and observed INP concentrations as a function of temperature. The black crosses
indicate INP measurements, and lines show model results from different parameterizations (Table 4).
Model grid points are selected at the same pressure levels and longitudes and latitudes as field
measurements.

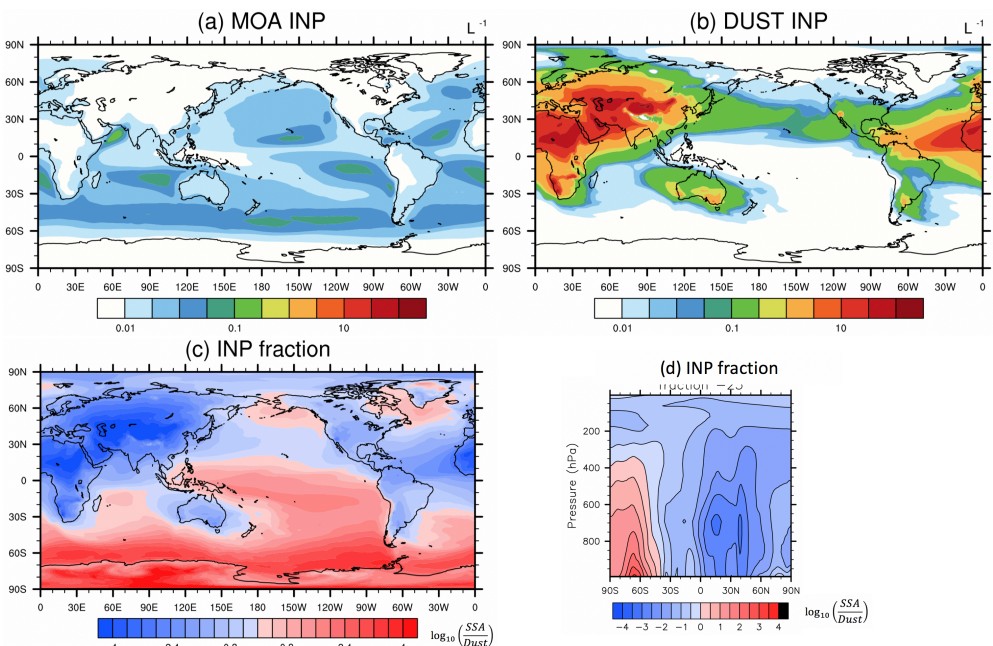

Figure 6. Spatial distribution of annual mean concentrations of (a) MOA INPs, (b) dust INPs, and (c) ratio of MOA INP concentration to dust INP concentration at 950 hPa, and (d) vertical cross sections of ratio of MOA INP concentration to dust INP concentration. INP concentrations are diagnosed at temperature of –25°C.



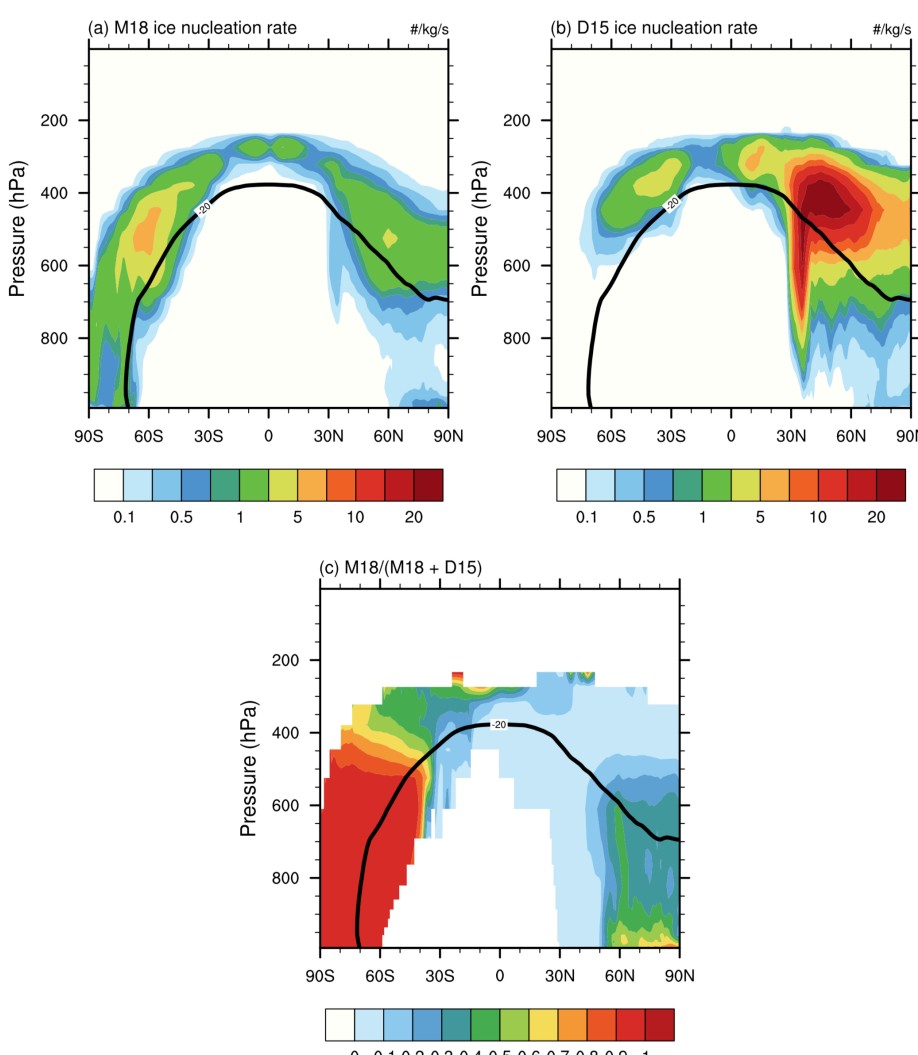

Figure 7. Annual zonal mean pressure-latitude cross sections of ice nucleation rates from (a) MOA, (b) dust,
and (c) MOA fraction of total ice production rate.



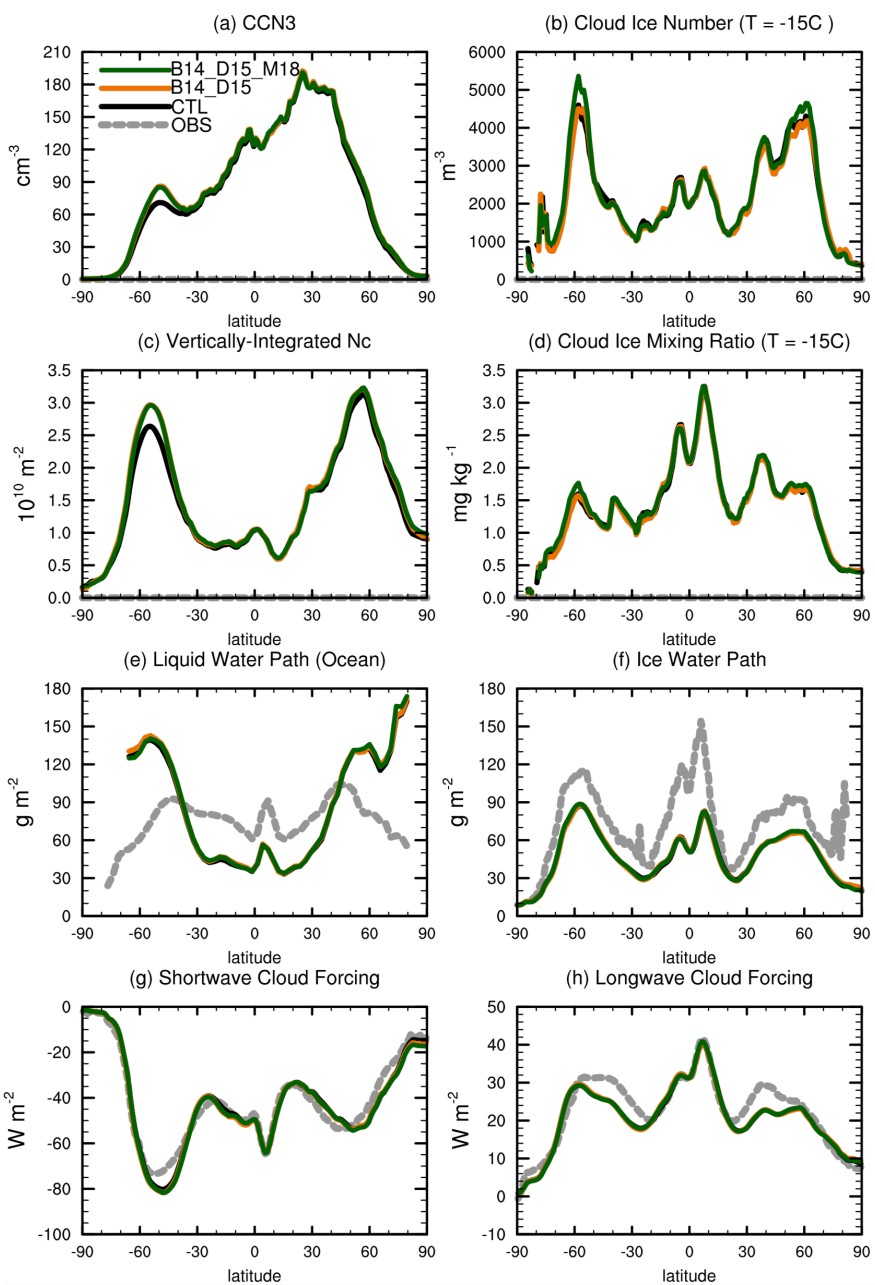

Figure 8. Annual zonal-mean distributions of (a) surface CCN concentration at S=0.1%, (b) cloud ice
number concentration on T=−15°C isotherm, (c) vertically-integrated cloud droplet number concentration,
(d) cloud ice mass mixing ratio on T= −15°C isotherm, (e) liquid water path over ocean, (f) ice water path,
(g) shortwave cloud forcing, and (h) longwave cloud forcing for CTL (black), B14_D15 (orange), and
B14_D15_M18 (green), along with available observations (gray dashed lines) as references.




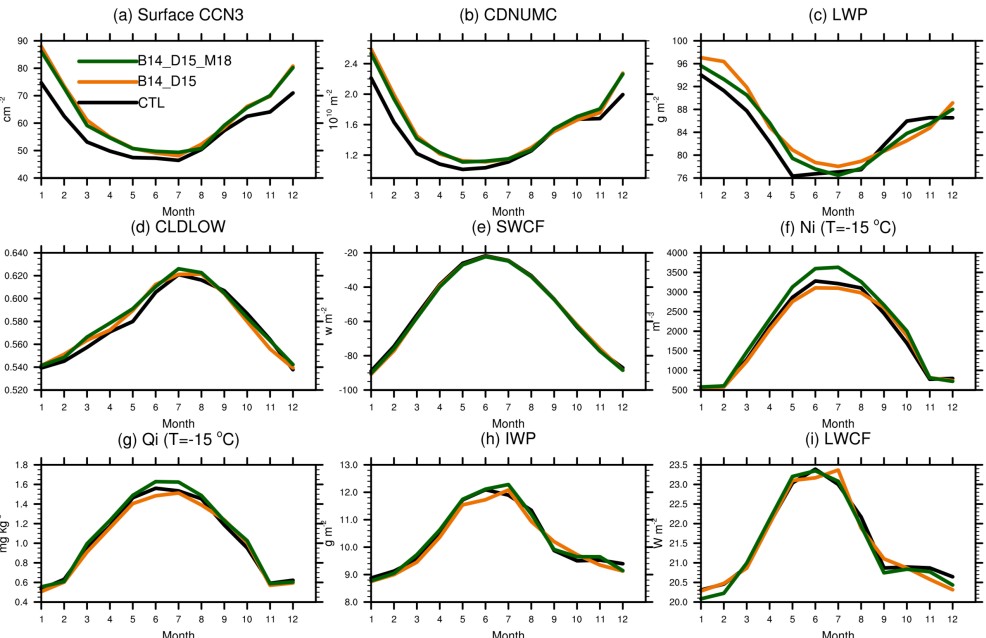

Figure 9. Seasonal cycle of (a) surface CCN at 0.1% supersaturation, (b) vertically-integrated cloud droplet
number concentration, (c) liquid water path, (d) low cloud amount, (e) shortwave cloud forcing, (f) cloud
ice number concentration on T= −15℃ isotherm, (g) cloud ice mass mixing ratio on T= −15℃ isotherm,
(h) ice water path (IWP), and (i) LWCF, for CTL (black), B14_D15 (orange) and B14_D15_M18 (green).