# Peer review of "Effects of Marine Organic Aerosols as Sources of Immersion-Mode Ice Nucleating Particles on High Latitude Mixed-Phase Clouds"

_Atmospheric Chemistry and Physics, 2020_

## Referee Comment (RC1) · Cyril Brunner (Referee) · 25 Sep 2020

General comments: The authors present a very interesting study on the effect of marine organic aerosol (MOA) on mixed-phase clouds. In particular, they present and validate results from three different MOA emissions schemes, quantify the resulting spatial cloud condensation nuclei and ice nucleating particles (INP) number concentrations and compare it to modeled INP concentrations of dust, using state of the art parametrizations. In contrast to previous work by other authors, they present data comparing the INP population of MOA to INPs of dust.

The writing (from an editorial standpoint) is to be commended. The methodology is

stringent and valid. The assumptions made are to a majority stated and their impact on the result comprehensibly assessed. The work addresses relevant scientific atmospheric questions with impacts on global climate simulations. The topic of the paper is well suited for ACP. I recommend the manuscript for publication if the following minor comments are addressed:

Specific comments: Page 2 (line 12/13). I do not fully understand why the three regions are stated. Are mixed-phase clouds only observed in the Arctic, Antarctic, and over the Southern Ocean? As the paper is also not focusing on these regions, I would propose to rephrase the sentence. E.g., include all regions or specify what is unique about mixed-phase clouds in the three stated regions.

Page 3 (lines 44-48). Kanji et al, 2017 provide an excellent overview of the different modes of freezing. The stated mechanism, however, was not introduced by Kanji et al., 2017. Please cite the original source or e.g. Vali, G., DeMott, P. J., Möhler, O., and Whale, T. F.: Technical Note: A proposal for ice nucleation terminology, Atmos. Chem. Phys., 15, 10263–10270, https://doi.org/10.5194/acp-15-10263-2015, 2015.

Page 15 (line 399). Is for the NULL approach the annual global MOA or sea salt emission 4.6 Tg yr-1? Please specify.

Page 23/24 (lines 652-683). The missing representation of secondary ice formation is nicely formulated. However, the study also does not model other IN species, such as ash, biomass-burning particles, or other land-borne bio particles. Please elaborate on the impact of this (valid) simplification on the study's results.

Page 32 (Table 1). Please explain variables in the caption, such as g, to support the reader's quicker understanding.

Page 46 (Figure 6). No need to change anything, just a general comment. With INP measurements, we are often divided about how the show the INP concentration most representatively over a long period. If we calculate the mean concentration, the result

will be "biased" towards higher INP concentrations if a few events with INP concentrations in the order of 102 to 103 are present. IMHO showing presenting the reader also with the median concentration provides a complete picture.

Supplement (Figure S1). The unit of plot d) is not fully visible.
* * *

---

## Referee Comment (RC2) · Anonymous Referee #2 · 29 Sep 2020

The manuscript titled: "Effects of Marine Organic Aerosols as Sources of Immersion-Mode Ice Nucleating Particles on High Latitude Mixed-Phase Clouds" discusses the impacts of adding marine organic aerosols (MOA) into the Community Atmosphere Model (version 6) on cloud properties. The study shows that introducing MOA as an aerosol species leads to a higher concentration of available CCN and INP, which results in different cloud properties. In contrast to the title of the study, the authors find that MOA has a much larger effect on the CCN concentration and related cloud radiative forcing, relative to INP and associated cloud radiative forcing. Nevertheless, the study does show that adding MOA increases the number of INPs and that MOA-INP are likely the most important INP species over the Southern Ocean, especially at heights below

400 hPa. I would like to commend the authors on a very well written and thorough study. However, I found the implementation of the emission scheme quite confusing and vague. I also have some additional comments below.

General comments:

In regards to the implementation of the MOA emission, it is not very clear how the MOA particles are handled. Firstly, how is the mixing state of the MOA determined (i.e. internally or externally mixed)? Does this just depend on the size mode of the emitted sea salt aerosol? If yes, how does the mass of the emitted MOA impact the resulting size of the sea salt and therefore the mixing state. Secondly, for the externally mixed MOA, how is the size versus number determined? Perhaps I am completely misunderstanding how this is done but please clarify in the methods.

When considering the internally mixed MOA with sea salt, is a freezing point depression considered when using the ice nucleation parametrizations? Based on the mass fractions of MOA relative to sea salt for the majority of the particles, sea salt appears to be the dominant component of the aerosol and I expect this to significantly lower the freezing efficiency of the MOA.

How does the lower hygroscopicity of the MOA relative to sea salt, factor into the available CCN numbers? Perhaps this will become clearer once the number and size of the MOA and its mixing state is explained more clearly.

A major point of the paper is that the addition of MOA greatly improves the representation of INP over the Southern Ocean and to some extent over the Arctic. However, when looking at the comparisons to the field observations, it could be argued that the N12 scheme does better at predicting observed INP concentrations. Therefore, perhaps it is better to compare the influence of adding MOA as an INP to the N12 scheme. I understand that this comparison will not be as straightforward but it seems a bit unfair to say that MOA is an important INP by comparing to D15, which clearly underestimates the observed INP concentrations and as the authors mention, only considers

dust particles larger than 500 nm. Nevertheless, I support the author's conclusion that MOA is likely an important INP species over remote regions such as the Southern Ocean. However, I think comparing to D15 may be making MOA out to be more important than it is.

As the Southern Ocean is where the largest changes in INP and CCN are observed, I find it quite surprising how little mention there is of sea ice extent. In theory, and to some degree is seems like some of the reported values in the study show this, the emission of MOA should be greatly reduced during the austral winter and early spring. In fact, I think the handling of sea ice in the model should be more clearly discussed and the seasonal influence on emission of MOA would warrant its own figure (perhaps in the supplement).

Minor comments:

Line 186-192: As I am not so familiar with MAM4, it is unclear to me where the SSA aerosol falls into one of the six aerosol species? Does it count as sea salt and are the cited parametrization used to determine the size distribution of sea salt?

Line 218-223: The G11 scheme requires the input of the chlorophyll concentration. This may show my ignorance but is chlorophyll predicted in the model or is it taken from fixed look up tables? If this comes from look up tables, does it account for seasonal variability and if so, how is the chlorophyll concentration over the Southern Ocean determined during the austral winter when satellite data for chlorophyll is limited to lower latitudes (below ∼55 degrees)?

Section 2.2.2: This section is a bit confusing as it is unclear which method was used here. Is the MOA emitted in the externally-mixed or internally-mixed approach? Based on the authors, it sounds like the Burrows et al, 2018 approach where the MOA is internally mixed should be used. Is that what is done here? If yes, and if the MOA is added to the sea salt fraction, does this not lead to an overall reduction in the hygroscopicity of the sea salt aerosols (or a freezing point depression)? Or is the increase in sea salt

mass due to the MOA fraction make the resulting particles large enough to overcome the reduction in hygroscopicity of the particles to act as CCN? Or does the MOA fraction also increase the number of sea salt+MOA particles. If the latter is the case how is the increase in number concentration handled as the MOA always scales with sea salt mass. Perhaps it would be worthwhile to clarify how the inclusion of the MOA fraction to the different size modes (i.e. Aitken, Accumulation, Coarse, Primary) acts to increase the number and size of the sea salt aerosol and lowers its overall hygroscopicity.

Table 1: Please explain the names of the aerosol species in MAM4. Perhaps these are standard in the modelling community but would be helpful for the reader to know what the acronyms stand for easily. That being said perhaps the same acronyms can be used in Table 1 and 2 for consistency and easy reference.

Section 2.2.2a: Please explain how the TOC is derived for the W15 scheme. Furthermore, is the TOC estimated from the sea surface or derived from the fraction emitted MOA? Later it is stated that it comes from the surface when expaliaing why W15 may be overestimating but please add that here.

Section 2.2.2b: Again here is it not immediately clear how this parametrization is implemented. Is this just dependent on temperature or is the MOA fraction somehow utilized here? Is the derived ns value applied to the total surface area and number concentration of the sea spray aerosol to activate a certain number of particles into ice crystals and if so what aerosol size modes are used? Also, how is the freezing point depression handled. Afterall, the particles are primarily composed of sea salt (at least the internally mixed ones)

Line 396-399: How do these studies justify such low fractions of MOA to sea salt when the laboratory studies report much higher fractions of MOA to SSA in the literature previously cited in the paper (ie. Prather et al, 2013 and Facchini et al, 2008) or are the large contributions of other compounds than MOA and sea salt in SSA? Furthermore, when looking at Figure 2, the fraction of MOA in SSA is much higher than the

values reported in this section. Granted Figure 2 appears in a region of high MOA emission, however it is unclear what SSA emission looks like globally. Perhaps it would be worthwhile to plot MOA/SSA emissions as an addition to Fig. S1.

Line 401-406: I find the switch between SSA and sea salt mass rather distracting when discussing the comparison of MOA fractions. Consider making this consistent as to my understanding, SSA is MOA+Sea salt and so it is just a different way of comparing the same values (e.g. MOA/(MOA+Sea salt) or MOA/Sea salt).

Table 5: Perhaps it would be worthwhile to also show the change in mean hygroscopicity of the emitted aerosols as the overall burdens do not seem to change much. Considering some of my comments on section 2.2.2, it would also be worthwhile to see the change in mean number and size of the sea salt and MOA aerosols emitted. I know this greatly varies by region but it would make it clearer to see how adding MOA to the model impacts the number of potential available particles to act as CCN. Also, as previously mentioned, based on the literature discussion, why is the fraction of MOA (MOA/Sea Salt) emission so much lower than reported values of around a few percent for particles larger than 1 micron and even higher for smaller particles?

Line 462: Consider revising to state that impact on clouds via CCN will be discussed next as the INP section follows section 3.2

Figure 3: Is there a reason why such a decrease is observed over the Tibetan plateau? Is there a way to add hatching for regions where the changes between simulations are significant or are these changes significant because they are the averages over the nine years?

Line 514-515: I would argue that the N12 parametrization does the best job of predicting the INP concentrations across the entire temperature range as shown in Figure 4. This is mischaracterized in these sentences. In fact based on the ability of N12, it would be possible to argue that including MOA emission is not needed to accurately predict the observed INP concentrations.

Line 521-523: Following up in the previous comment, the fact that combining M18 and D15 still under predicts, shows that the MOA addition is not as good as just using the N12 parametrization. And one could argue that when using the entire size distribution for dust nucleation, the majority of INPs would accurately simulated.

Figure 5: How are the parametrizations drawn for the flight campaigns (e.g. Socrates), where INP measured aboard the aircraft are sometimes collected at different altitudes?

Line 540-551 and Figure 6: Why was INPs at -25 C and 950 hPa used for this analysis here? For cloud formation, this seems highly irrelevant as it is rarely -25 C at 950 hPa especially in the regions where MOA is expected to be important (over ice-free regions of the Ocean). In fact, how common is it for MPCs to exist at this height. This seems like an unfair height for showing the importance of MOA as INPs as it is a height where MOA concentrations are extremely high due to being within the boundary layer and at a temperature where the ability of MOA to freeze is essentially maximized based on field measurement techniques used at temps above ∼-30 C.

Line 552-561: I generally agree with the explanation for the observed differences shown in Fig.6d. However, how does the model handle the sea ice coverage over the Southern Ocean during Austral winter? As the sea ice extent should extend as far north as approximately 60 S. Therefore, it important to know how the model handles the emission of MOA during the austral winter and spring months, especially as the sea ice extent is prescribed based on climatology (see lines 354-356). Does this mean that the entire year assumes a constant sea ice extent or does it change based on season. Depending on how this is handled, it could have large implications for both the INP and CCN distribution due to MOA emissions.

Figure 7: How does the model produce ice nucleation or even MOA at such high latitudes in the Southern Hemisphere at the surface when the center of the Antarctic ice sheet is ∼3000 m (700 hPa)?

Line 579-581: Mentioning seasonal dependence is quite interesting especially concerning my previous comment about the sea-ice extent. Therefore, I think it would be worthwhile to show how the emission of MOA changes over the Southern Ocean between austral summer and winter and how this influences the freezing rates.

Line 590-592: What does this increase in percent mean? Can you report the change in the number of CDNUMC? Also the fact that there is a difference between austral summer and winter might point to a change in the sea ice extent in the model as well as biological activity.

Line 599: why do you switch to an isotherm of -15 now? Previously the -25 isotherm was used and the figures also have the -20 isotherm. Perhaps it is better to be consistent and choose one isotherm throughout or at least explain why different isotherms are chosen.

Line 628-630: Could this also be due to sea ice extent?

Line 655-661: Discussion on the differences between bubble bursting (which is implemented in the model) and jet drops (which is not) does not seem necessary. Perhaps it is just fine to just mention that more observations/ fundamental understanding are needed for implementation of jet drops as is not clear why the differences in size of the emitted aerosols matter here. If this is important, please expand on why that could make a significant difference to the observed results and overall importance of MOA.

Editorial comments:

Line 47: "replace to" with "and" as the sentence should read: "temperature is between -38 and 0..."

Line 49: Please consider rephrasing the sentence to read: "INPs have different characteristics depending on their composition and origin" as it does not make sense as it is written.

Line 52: Consider adding some citations when you mention the uncertainty in the ability of black carbon to act as INPs. To my understanding, the evidence is mounting that BC

is irrelevant in the MPC region.

Line 74: Please add Ault et al, 2013 to the reference list.

Line 123: Please add "method" or something similar after: "[Chl-a]-based"

Figure 1: Fix unit for micro

Figure 3: The 0.1 % supersaturation is not showing up well. Also, the longitude representation (248 W) seems a bit odd, but perhaps that's just a personal preference. In the Figure caption, it might be nice to state that percent change in surface CCN comes from comparing B14-BASE.

Figure 4: Consider flipping the color bar so that the warmest temperatures are at the top and the coldest at the bottom.

Figure 6: Please fix panel d to be consistent with the other panels.

---

## Author Response (AR1)

**Response to Dr. Cyril Brunner**

We thank Dr. Brunner for his careful reading and review of our paper. Our detailed responses to his comments follow. Reviewer comments are in blue, our responses are in black, and our corresponding revisions in the manuscript are in red.

**General comments:**

The authors present a very interesting study on the effect of marine organic aerosol (MOA) on mixed-phase clouds. In particular, they present and validate results from three different MOA emissions schemes, quantify the resulting spatial cloud condensation nuclei and ice nucleating particles (INP) number concentrations and compare it to modeled INP concentrations of dust, using state of the art parametrizations. In contrast to previous work by other authors, they present data comparing the INP population of MOA to INPs of dust.

The writing (from an editorial standpoint) is to be commended. The methodology is stringent and valid. The assumptions made are to a majority stated and their impact on the result comprehensibly assessed. The work addresses relevant scientific atmospheric questions with impacts on global climate simulations. The topic of the paper is well suited for ACP. I recommend the manuscript for publication if the following minor comments are addressed:

We thank the reviewer for the encouraging comments. We have revised the manuscript following your comments and clarified the text to improve the paper.

**Specific comments:**

Page 2 (line 12/13). I do not fully understand why the three regions are stated. Are mixed-phase clouds only observed in the Arctic, Antarctic, and over the Southern Ocean? As the paper is also not focusing on these regions, I would propose to rephrase the sentence. E.g., include all regions or specify what is unique about mixed-phase clouds in the three stated regions.

We thank the reviewer for the suggestion. We have revised the sentence to read:

"Mixed-phase clouds are frequently observed in high-latitude regions, and..."

Page 3 (lines 44-48). Kanji et al, 2017 provide an excellent overview of the different modes of freezing. The stated mechanism, however, was not introduced by Kanji et al., 2017. Please cite the original source or e.g. Vali, G., DeMott, P. J., Möhler, O., andWhale, T. F.: Technical Note: A proposal for ice nucleation terminology, Atmos. Chem.Phys., 15, 10263–10270, https://doi.org/10.5194/acp-15-10263-2015, 2015.

We thank the reviewer for the suggestion. We have added the reference as you suggested, and the
revised sentence reads as:

"In mixed-phase clouds in which air temperature is between $-38°$ C and $0°$ C, ice is initialized only by
heterogeneous nucleation on ice nucleating particles (INPs) (Vali et al., 2015)."

Page 15 (line 399). Is for the NULL approach the annual global MOA or sea salt emission 4.6 Tg yr-1?
Please specify.

We thank the reviewer for the suggestion. We meant the annal global MOA emission of 4.6 Tg $yr^{-1}$.
We have revised the sentence as:

"The NULL approach only gives an annual global MOA emission of 4.6 Tg $yr^{-1}$".

Page 23/24 (lines 652-683). The missing representation of secondary ice formation is nicely
formulated.   However, the study also does not model other IN species, such as ash, biomass-burning
particles, or other land-borne bio particles. Please elaborate on the impact of this (valid) simplification
on the study's results.

We thank the reviewer for the excellent comment. We have added some discussion on missing
representation of other types of INPs in the model in the newly added third paragraph of section 4.
The revised sentences read as:

"In this study, other potential INP species than dust and MOA, such as ash, biomass-burning particles,
or other land-borne biological particles (Hoose et al., 2010; Jahn et al., 2020; Schill et al., 2020) are
not represented in the model. These INP species can be regionally important at certain temperature
regimes of mixed-phase clouds. Accounting for these species may increase the INP concentrations
predicted in the model and change the mixed-phase cloud properties, particularly at warmer
temperatures > -15ºC. The impacts of these INP species will be quantified in our future studies."

Page 32 (Table 1).   Please explain variables in the caption, such as g, to support the reader's quicker
understanding.

We thank the reviewer for the suggestion. We have added the explanations for variables in Table 1,
and the revised caption for Table 1 reads:

Table 1. Aerosol species in MAM4 modes

| | Accumulation | Aitken | Coarse | Primary Carbon |
|---|---|---|---|---|
| Species[1] | num_a1, so4_a1, pom_a1, soa_a1, bc_a1, dst_a1, ncl_a1, moa_a1 | num_a2, so4_a2, soa_a2, ncl_a2, dst_a2, moa_a2 | num_a3, dst_a3, ncl_a3, so4_a3 | num_a4, pom_a4, bc_a4, (moa_a4 if externally mixed) |
| Size range | $0.08 - 1$ μm | $0.02 - 0.08$ μm | $1-10$ μm | $0.08 - 1$ μm |
| Standard Deviation $\sigma g$ | 1.6 | 1.6 | 1.2 | 1.6 |
| Number-median diameter $Dgn$ | $1.1 \times 10^{-7}$ | $2.6 \times 10^{-8}$ | $2.0 \times 10^{-6}$ | $5.0 \times 10^{-8}$ |
| Low bound $Dgn$ | $5.35 \times 10^{-8}$ | $8.7 \times 10^{-9}$ | $4.0 \times 10^{-7}$ | $1.0 \times 10^{-8}$ |
| High bound $Dgn$ | $4.8 \times 10^{-7}$ | $5.2 \times 10^{-8}$ | $4.0 \times 10^{-5}$ | $1.0 \times 10^{-7}$ |

[1]so4_aX: sulfate mass mixing ratio in mode X; pom_aX: particulate organic matter (POM) mass
mixing ratio in mode X; soa_aX: secondary organic aerosol (SOA) mass mixing ratio in mode X;
bc_aX: black carbon (BC) mass mixing ratio in mode X; dst_aX: dust mass mixing ratio in mode X;
ncl_aX: sea salt mass mixing ratio in mode X; moa_aX: marine organic aerosol (MOA) mass mixing
ratio in mode X; and num_aX: number mixing ratio of mode X. *_a1: accumulation mode; *_a2:
Aitken mode; *_a3: coarse mode; and *_a4: coarse mode.

Page 46 (Figure 6). No need to change anything, just a general comment. With INP measurements, we
are often divided about how the show the INP concentration most representatively over a long period.
If we calculate the mean concentration, the result will be "biased" towards higher INP concentrations
if a few events with INP concentrations in the order of 102 to 103 are present. IMHO showing
presenting the reader also with the median concentration provides a complete picture.

We thank the reviewer for the suggestion. We agree with the reviewer that mean concentration will be
"biased" towards the higher INP. However, in GCMs we are simulating the climatological states of
aerosols, not episodical events.

Supplement (Figure S1). The unit of plot d) is not fully visible

Thanks. We have fixed this. The revised Figure looks like:

[Figure]

**Response to reviewer 2**

We thank the anonymous reviewer for the careful review and constructive comments on our manuscript. Our responses to individual comments follow. Reviewer comments are in blue, our responses are in black, and our corresponding revisions in the manuscript are in red.

The manuscript titled: "Effects of Marine Organic Aerosols as Sources of Immersion Mode Ice Nucleating Particles on High Latitude Mixed-Phase Clouds" discusses the impacts of adding marine organic aerosols (MOA) into the Community Atmosphere Model (version 6) on cloud properties. The study shows that introducing MOA as an aerosol species leads to a higher concentration of available CCN and INP, which results in different cloud properties. In contrast to the title of the study, the authors find that MOA has a much larger effect on the CCN concentration and related cloud radiative forcing, relative to INP and associated cloud radiative forcing. Nevertheless, the study does show that adding MOA increases the number of INPs and that MOA-INP are likely the most important INP species over the Southern Ocean, especially at heights below 400 hPa. I would like to commend the authors on a very well written and thorough study. However, I found the implementation of the emission scheme quite confusing and vague. I also have some additional comments below.

We thank the reviewer for the constructive comments, which greatly improve the clarity of our paper. In this study, as the title indicates, we focus on the MOA INP effects, since earlier studies (Meskhidze et al., 2011; Gantt et al., 2012; Burrow et al., 2018), using earlier versions of CAM, have investigated the MOA CCN effects. As the reviewer correctly indicates, INP effects of MOA on cloud properties and radiative forcing can be regionally strong in remote marine environments such as over the Southern Ocean.

We have significantly revised the method section of our manuscript to improve its clarity related to the emission scheme.

**General comments:**

In regards to the implementation of the MOA emission, it is not very clear how the MOA particles are handled. Firstly, how is the mixing state of the MOA determined (i.e. internally or externally mixed)? Does this just depend on the size mode of the emitted sea salt aerosol? If yes, how does the mass of the emitted MOA impact the resulting size of the sea salt and therefore the mixing state. Secondly, for the externally mixed MOA, how is the size versus number determined? Perhaps I am completely misunderstanding how this is done but please clarify in the methods.

We thank the reviewer for the suggestion. In the following, we will clarify the confusing points one by one.

Firstly, regarding the mixing state of the MOA: MAM in CAM6 adopts the modal approach, where aerosol species are assumed to be internally mixed within a mode, and externally mixed between modes. MOA is emitted into the fine aerosol modes with different assumptions of mixing state with inorganic sea salt: (1) MOA is emitted into the Aitken and accumulation modes together with sea salt in the case of internally mixed with sea salt; or (2) MOA is emitted into the Aitken and primary carbon mode separately from sea salt in the case of externally mixed with sea salt. MOA is not emitted into the coarse mode though sea salt does. In addition, there is another assumption of whether the experimentally derived parameterizations of sea spray aerosol mass emission flux represent the total emission of MOA and sea salt or only account for the emission of sea salt. In the former case, MOA will *replace* the mass and number emission fluxes of sea salt. In the latter case, MOA will *add* onto both the sea salt mass and number emission fluxes. Burrows et al. (2018) tested different combinations of the two assumptions and found that the "internally-mixed" and "added" approach for MOA provides the most physically realistic configuration compared to the observations. Thus, we used this configuration in our study. In this configuration, the emission of MOA will not impact the emission fluxes of sea salt. We acknowledge that current experiments and observations do not provide precise constrains on the mixing state. For the impacts of different assumption of mixing state we refer the readers to Burrows et al. (2018).

Second, the emitted MOA mass flux is calculated as:

$$F_{MOA/SSA} = \frac{M_{MOA}}{M_{sea\ spray}} = \frac{M_{MOA}}{M_{MOA} + M_{sea\ salt}}$$

where $F_{MOA/SSA}$ is the mass fraction of MOA in total SSA. G11, B14, and NULL emission schemes are used to calculate $F_{MOA/SSA}$, respectively. $M_{sea\ salt}$ is the emitted sea salt mass, calculated following the parameterization of Mårtensson et al. (2003) for dry particle diameters from 0.020 to 2.8 µm, and Monahan et al. (1986) from 2.8 to 10 µm in the model.

Third, the emitted MOA number flux is calculated based on the emitted MOA mass flux for a given particle diameter within the emission size range (from 0.020 to 2.8 µm) of the Mårtensson et al. parameterization, and the particle density of MOA, the latter of which is set to be 1601 kg m$^{-3}$, as given in Table 2.

In response to the comments made by reviewers, we revised the manuscript:

We added some discussions on the mixing state of aerosol in Section 2.1:

"MAM in CAM6 adopts the modal approach, where aerosol species are assumed to be internally mixed within a mode, and externally mixed between modes. MOA is emitted into the fine aerosol modes with different assumptions of mixing state with inorganic sea salt: (1) MOA is emitted into the Aitken and accumulation modes together with sea salt in the case of internally mixed with sea salt; or (2) MOA is emitted into the Aitken and primary carbon mode separately from sea salt in the case of externally mixed with sea salt. In addition, there is another assumption of whether the experimentally derived parameterizations of SSA mass emission flux represent the total emission of MOA and sea salt or only account for the emission of sea salt. In the former case, MOA will *replace* the mass and number emission fluxes of sea salt. In the latter case, MOA will *add* onto the sea salt mass and number emission fluxes. Burrows et al. (2018) tested different combinations of the two assumptions and found that the "internally-mixed" and "added" MOA approach provides the most physically realistic configuration compared to the observations. Thus, in our study we use this configuration but
acknowledge that current observations do not provide precise constrains on the mixing state."

The discussion of emitted MOA number mixing ratio is added in Section 2.2.1 of the revised
manuscript:

"The MOA number emission flux is calculated based on the MOA mass emission flux for a given
particle diameter within the emission size range (from 0.020 to 2.8 µm for the Mårtensson et al.
parameterization) and particle density of MOA, the latter of which is set to be 1601 kg m$^{-3}$ (Liu et al.,
2012), as given in Table 2."

When considering the internally mixed MOA with sea salt, is a freezing point depression considered
when using the ice nucleation parametrizations? Based on the mass fractions of MOA relative to sea
salt for the majority of the particles, sea salt appears to be the dominant component of the aerosol and
I expect this to significantly lower the freezing efficiency of the MOA.

We thank the reviewer for the good suggestion. The M18 ice nucleation scheme is derived based on
the correlation between ambient sea spray aerosols and INPs measured by the Count-Flow Diffusion
Chamber (CFDC) during the "clean scenario" at Mace Head. This means that this parameterization
has already accounted for the freezing point depression effect when MOA INPs in droplets induce
freezing in CFDC. The W15 scheme is developed based on laboratory measurements of
immersion-freezing of materials aerosolized from sea surface microlayer samples collected in the N.
Atlantic and Arctic Oceans. Thus, this parameterization should have also accounted for the freezing
point depression effect during the freezing of droplets induced by MOA.

How does the lower hygroscopicity of the MOA relative to sea salt, factor into the available CCN
numbers? Perhaps this will become clearer once the number and size of the MOA and its mixing state
is explained more clearly.

We thank the reviewer for this question. The hygroscopicity of the MOA is 0.1, compared with 1.16
of sea salt, as listed in Table 2. In our reply to your comment above, MOA is assumed to be internally
mixed with sea salt in the accumulation and Aitken modes in this study. The hygroscopicity of
aerosols in a mode is calculated based on volume-averaged hygroscopicities of all aerosol species in
the mode and used in the aerosol activation calculation. After considering MOA in the model, the
mode volume-averaged hygroscopicity is reduced compared to pure sea salt aerosol. However,
number concentrations of CCN are still increased with the "added" MOA into the model.

We added the following sentence in Section 2.2 of the revised manuscript:

"The mode volume-averaged hygroscopicity is reduced due to lower hygroscopicity of MOA.
However, based on the method to calculate sea salt emission (Liu et al., 2012) for a given aerosol
mode, the "added" MOA mass increases the number concentrations of aerosols in the Aitken and
accumulation modes, which overcomes the reduction in mode hygroscopicity to activate more CCN."

A major point of the paper is that the addition of MOA greatly improves the representation of INP
over the Southern Ocean and to some extent over the Arctic. However, when looking at the
comparisons to the field observations, it could be argued that the N12 scheme does better at predicting
observed INP concentrations. Therefore, perhaps it is better to compare the influence of adding MOA
as an INP to the N12 scheme. I understand that this comparison will not be as straightforward but it
seems a bit unfair to say that MOA is an important INP by comparing to D15, which clearly
underestimates the observed INP concentrations and as the authors mention, only considers dust
particles larger than 500 nm. Nevertheless, I support the author's conclusion that MOA is likely an
important INP species over remote regions such as the Southern Ocean. However, I think comparing
to D15 may be making MOA out to be more important than it is.

We thank the reviewer for the good comment. It seems that the N12 scheme has the better
performance than D15 in Figure 4. However, the field campaigns used in Figure 4 are marine aerosol
dominant/contained scenario campaigns. MOA is identified as an important INP source during these
campaigns from measurements (McCluskey, Ovadnevaite, Rinaldi, et al., 2018b; McCluskey, Hill,
Humphries, et al., 2018a). For example, CAPRICORN and SOCRATES, these two campaigns were
conducted over the Southern Ocean where dust aerosol was less influenced. The "clean scenario"
(McCluskey, Ovadnevaite, Rinaldi, et al., 2018b) during the Mace Head campaign is selected to focus
on marine aerosol influence. Thus, dust should not be expected to be the dominant INPs as indicated
by the N12 scheme which only considers dust INPs. This suggests that N12 may overestimate dust
INPs. This is also confirmed in our previous study (Shi and Liu, 2019) which compared the D15 and
N12 schemes with the field observations conducted in the Arctic subject to the major influence from
dust. It was found that D15 underestimates the observed INPs while N12 has a better performance.
However, the host aerosol-climate model was found to significantly underestimate the observed dust
concentrations by up to a factor of 10 due to missing local Arctic sources and too weak transport of
dust from low latitudes. Considering the low bias of dust in the Arctic predicted by the host model,
that study suggested that D15 overall has the better performance in representing dust INPs.

To improve the clarity, we revised the experiment description in section 2.3 of the revised manuscript
as

"The control experiment (CTL) is the same as BASE except that the D15 dust ice nucleation scheme
was used to replace the CNT scheme in BASE, because D15 gave a better model performance
compared with observations in our previous study (Shi and Liu, 2019)."

We added in section 3.3 of the revised manuscript:

"The N12 scheme has the better performance than D15 in Figure 4. However, the field campaigns used in Figure 4 are marine aerosol dominant/contained scenario campaigns. MOA is identified as an important INP source during these campaigns from measurements (McCluskey, Ovadnevaite, Rinaldi, et al., 2018b; McCluskey, Hill, Humphries, et al., 2018a). Thus, dust should not be expected to be the dominant INPs as indicated by the N12 scheme which only considers dust INPs. This suggests that N12 may overestimate dust INPs, which is consistent with our earlier study (Shi and Liu, 2019)."

As the Southern Ocean is where the largest changes in INP and CCN are observed, I find it quite surprising how little mention there is of sea ice extent. In theory, and to some degree is seems like some of the reported values in the study show this, the emission of MOA should be greatly reduced during the austral winter and early spring. In fact, I think the handling of sea ice in the model should be more clearly discussed and the seasonal influence on emission of MOA would warrant its own figure (perhaps in the supplement).

We agree with the reviewer that sea ice has a significant influence on the emission of MOA. As shown in Figure 2a, MOA concentrations are higher in January (austral summer) and lower in July (austral winter) at Amsterdam Island in the Southern Hemisphere, which reflects the seasonal change of MOA emissions.

In this study, all experiment is set up using "F2000climo" component in CESM2-CAM6 model. As described in section 2.3, "All simulations were performed for 10 years with prescribed climatological sea surface temperatures and sea ice." The atmosphere model is not coupled with the sea ice model, but uses the prescribed climatological sea ice as its boundary condition. The sea ice data has a seasonal variation, namely, it has 12 months as a time dimension. The sea ice extent will impact the seasonal variation of MOA emission.

Following the reviewer's comment, we added a figure in the supplement showing the seasonal variation of sea ice extent and related discussion on the impact of sea ice extent on the emission of MOA in the revised manuscript in section 3.1:

 "The sea ice extent prescribed in the model as a boundary condition has a strong seasonal variation over the Southern Ocean, as shown in supplementary Figure S2. This can greatly impact the emission of MOA there (e.g., low emission during the austral winter and early spring)."

[Figure]

Figure S2. Seasonal variation of global sea ice extent, shown as sea ice fraction in 12 months.

**Minor comments:**

Line 186-192: As I am not so familiar with MAM4, it is unclear to me where the SSA aerosol falls
into one of the six aerosol species? Does it count as sea salt and are the cited parametrization used to
determine the size distribution of sea salt?

In this study, the sea spray aerosol (SSA) refers to sea salt plus MOA. Sea salt is part of SSA. MOA is
added into MAM4 as a new species (sea salt is already implemented in MAM4), which means newly
predicted variables in MAM4 to trace the temporal and spatial variations of MOA mass mixing ratios.
In the "internally-mixed" and "added" MOA approach used in this study, the emitted mass mixing
ratio of MOA is dependent on the emission of sea salt mass mixing ratio and the mass fraction of
MOA in total SSA, $F_{MOA/SSA}$. The Mårtensson et al. (2003) and Monahan et al. (1986)
parameterizations are used to calculate the sea salt emission flux, while the G11, B14, and NULL
emission schemes are used to calculate $F_{MOA/SSA}$.

Line 218-223: The G11 scheme requires the input of the chlorophyll concentration. This may show
my ignorance but is chlorophyll predicted in the model or is it taken from fixed look up tables? If this
comes from look up tables, does it account for seasonal variability and if so, how is the chlorophyll concentration over the Southern Ocean determined during the austral winter when satellite data for
chlorophyll is limited to lower latitudes (below ~55 degrees)?

Thank the reviewer for the comment. We used the prescribed climatological data for chlorophyll
concentrations, and this data has a seasonal variation, namely, it has 12 months as the time dimension.
Below we plotted the chlorophyll concentrations we used in the model for each month, shown as
Figure R1. We noticed much larger values during the austral summer (DJF) than those during the
austral winter (JJA) over the Southern Ocean. This is expected since phytoplankton activities are
minimal in the austral winter in the Southern Hemisphere high latitudes.

[Figure]

Figure R1. Seasonal variation for global distribution of chlorophyll concentrations.

Section 2.2.2: This section is a bit confusing as it is unclear which method was used here. Is the MOA
emitted in the externally-mixed or internally-mixed approach? Based on the authors, it sounds like the
Burrows et al, 2018 approach where the MOA is internally mixed should be used. Is that what is done
here? If yes, and if the MOA is added to the sea salt fraction, does this not lead to an overall reduction
in the hygroscopicity of the sea salt aerosols (or a freezing point depression)? Or is the increase in sea
salt mass due to the MOA fraction make the resulting particles large enough to overcome the
reduction in hygroscopicity of the particles to act as CCN? Or does the MOA fraction also increase the
number of sea salt+MOA particles. If the latter is the case how is the increase in number concentration handled as the MOA always scales with sea salt mass. Perhaps it would be worthwhile to clarify how
the inclusion of the MOA fraction to the different size modes (i.e. Aitken, Accumulation,
Primary) acts to increase the number and size of the sea salt aerosol and lowers its overall
hygroscopicity.

We thank the reviewer for the comments. Following the suggestion of Burrows et al. (2018), we used
the "internally-mixed" and "added" approach for MOA in this study. We have made it clear in the
revised manuscript. In this "internally-mixed approach" MOA is emitted into the Aitken and
accumulation modes along with sea salt.

Yes, the mode averaged hygroscopicity is reduced due to lower hygroscopicity of MOA. However, the
"added" MOA mass increases the number concentrations of sea salt+MOA aerosols in the size ranges
of the Aitken and accumulation modes, which overcomes the reduction in mode hygroscopicity to
activate more CCN.

As documented in Liu et al. (2012), when calculating sea salt emission, sea salt size distribution is
divided into many small size bins, where the emission of sea salt mass mixing ratio is calculated based
on the Mårtensson et al. (2003) and Monahan et al. (1986) parameterizations, and the emission of sea
salt number mixing ratio for a given size bin is derived based on the mass emission. The total mass
and number emissions for an aerosol mode are summed up over the relevant size bins. In this study,
the "added" MOA number mixing ratio is calculated similarly based on the emitted MOA mass
mixing ratio, particle size for a given bin, and the particle density of MOA, as number is proportional
to mass for a given size bin. Therefore, after considering MOA in the model, number concentrations
of sea salt+MOA in the Aitken and accumulation modes are increased.

To avoid confusion, we revised the MOA emission description in section 2.2.2 of the revised
manuscript as

"MOA is emitted into different aerosol modes depending on mixing state of MOA and sea salt
(Burrows et al., 2014, 2018). In the internally-mixed emission approach, MOA is emitted into the
accumulation and Aitken modes along with sea salt, as shown in Table 1. In contrast, MOA is emitted
into the Aitken and primary carbon modes in the externally-mixed emission approach. Furthermore,
the emission of MOA can replace or be added to sea salt emission in terms of mass and number in the
model. Burrows et al. (2018) found that simulated MOA amounts, seasonal cycles, and impacts on
CCN over the Southern Ocean show better agreement with observations under the assumption that
emitted MOA is added to, and internally mixed with sea salt. Thus, we used the "internally-mixed"
and "added" approach for MOA emission in this study. As shown in Table 2, the hygroscopicity of
MOA is set to be 0.1 following Burrows et al. (2014, 2018), compared to 1.16 for sea salt. The mode
hygroscopicity is calculated as the volume-weighted average of hygroscopicities of all species in a
mode, which is then used in the Abdul-Razzak and Ghan (2000) droplet activation parameterization in
CAM6. The mode hygroscopicity is reduced due to lower hygroscopicity of MOA. However, based on
the method to calculate sea salt emission (Liu et al., 2012) for a given aerosol mode, the "added"
MOA mass increases the number concentrations of particles in the Aitken and accumulation modes,
which overcomes the reduction in mode hygroscopicity to activate more CCN."

Table 1: Please explain the names of the aerosol species in MAM4. Perhaps these are standard in the modelling community but would be helpful for the reader to know what the acronyms stand for easily. That being said perhaps the same acronyms can be used in Table 1 and 2 for consistency and easy reference.

Thanks, we have added the explanation of the names of the aerosol species in MAM4. Please note that we use *_aX to indicate aerosol species in mode X (i.e., a1 for accumulation, a2 for Aitken, a3 for coarse, and a4 for primary carbon mode.

Table 1. Aerosol species in MAM4 modes

| | Accumulation | Aitken | Coarse | Primary Carbon |
|---|---|---|---|---|
| Species[1] | num_a1, so4_a1, pom_a1, soa_a1, bc_a1, dst_a1, ncl_a1, moa_a1 | num_a2, so4_a2, soa_a2, ncl_a2, dst_a2, moa_a2 | num_a3, dst_a3, ncl_a3, so4_a3 | num_a4, pom_a4, bc_a4, (moa_a4 if externally mixed) |
| Size range | $0.08 - 1$ μm | $0.02 - 0.08$ μm | $1-10$ μm | $0.08 - 1$ μm |
| Standard Deviation $\sigma g$ | 1.6 | 1.6 | 1.2 | 1.6 |
| Number-median diameter $Dgn$ | $1.1 \times 10^{-7}$ | $2.6 \times 10^{-8}$ | $2.0 \times 10^{-6}$ | $5.0 \times 10^{-8}$ |
| Low bound $Dgn$ | $5.35 \times 10^{-8}$ | $8.7 \times 10^{-9}$ | $4.0 \times 10^{-7}$ | $1.0 \times 10^{-8}$ |
| High bound $Dgn$ | $4.8 \times 10^{-7}$ | $5.2 \times 10^{-8}$ | $4.0 \times 10^{-5}$ | $1.0 \times 10^{-7}$ |

[1]so4_aX: sulfate mass mixing ratio in mode X; pom_aX: particulate organic matter (POM) mass mixing ratio in mode X; soa_aX: secondary organic aerosol (SOA) mass mixing ratio in mode X; bc_aX: black carbon (BC) mass mixing ratio in mode X; dst_aX: dust mass mixing ratio in mode X; ncl_aX: sea salt mass mixing ratio in mode X; moa_aX: marine organic aerosol (MOA) mass mixing ratio in mode X; and num_aX: number mixing ratio of mode X. *_a1: accumulation mode; *_a2: Aitken mode; *_a3: coarse mode; and *_a4: coarse mode.

Table 2. Aerosol species and physical properties

| Species | Name | Density (kg m$^{-3}$) | Hygroscopicity |
|---|---|---|---|
| BC | Black carbon | 1700 | $1.0 \times 10^{-10}$ |
| SO4 | Sulfate | 1770 | 0.507 |
| SOA | Secondary organic | 1000 | 0.14 |
| POA | Primary organic | 1000 | $1.0 \times 10^{-10}$ |
| DST | Dust | 2600 | 0.068 |
| NCL | Sea salt | 1900 | 1.16 |
| MOA | Marine organic aerosol | 1601 | 0.1 |

Section 2.2.2a: Please explain how the TOC is derived for the W15 scheme. Furthermore, is the TOC
estimated from the sea surface or derived from the fraction emitted MOA? Later it is stated that it
comes from the surface when expalianing why W15 may be overestimating but please add that here.

We thank the reviewer for the suggestion. Yes, the W15 scheme is derived based on TOC in sea
surface microlayer samples, which may not be representative of ambient MOA. We added a note in
the description of W15 in section 2.2.2a as

"W15 is developed based on the TOC in the sea surface microlayer samples, which may not be
representative of ambient MOA."

Section 2.2.2b: Again here is it not immediately clear how this parametrization is implemented. Is this
just dependent on temperature or is the MOA fraction somehow utilized here? Is the derived ns value
applied to the total surface area and number concentration of the sea spray aerosol to activate a certain
number of particles into ice crystals and if so what aerosol size modes are used? Also, how is the
freezing point depression handled. Afterall, the particles are primarily composed of sea salt (at least
the internally mixed ones)

We thank the reviewer for the suggestion. $n_s(T)$ in Equation (8) is used to calculate MOA INPs
based on

$$N_{INP}(T) = N_{tot} S_{ae} n_s(T)$$

where $S_{ae}$ and $N_{tot}$ are the total surface area and number mixing ratio of SSA, calculated for the
Aitken and accumulation modes, respectively. M18 is derived based on the correlation between
ambient SSA aerosols and INPs during the "clean scenario" at Mace Head Station in August 2015.
INPs were measured by CFDC, and thus, should have accounted for the effect of freezing point
depression on droplets freezing.

We modified the description of M18 in the section 2.2.2b as

"MOA INP number concentration is then calculated by: $N_{INP}(T) = N_{tot} S_{ae} n_s(T)$, where $S_{ae}$ and
$N_{tot}$ are the total surface area and number mixing ratio of SSA, calculated for the Aitken and
accumulation modes, respectively."

Line 396-399: How do these studies justify such low fractions of MOA to sea salt when the laboratory
studies report much higher fractions of MOA to SSA in the literature previously cited in the paper (ie.
Prather et al, 2013 and Facchini et al, 2008) or are the large contributions of other compounds than
MOA and sea salt in SSA? Furthermore, when looking at Figure 2, the fraction of MOA in SSA is
much   of high MOA emission, however it is unclear what SSA emission looks like globally. Perhaps
it would be worthwhile to plot MOA/SSA emissions as an addition to Fig. S1.

We thank the reviewer for the comments. The ratios of MOA emission to sea salt emission include the
emission of sea salt in the coarse mode, which dominates the total sea salt emission. Figure 2c shows the simulated and measured mass fraction of MOA in SSA for the Aitken and accumulation modes (MOA is not considered in the coarse mode). We added a note in section 3.1 to make it clearer when we talk about the ratios of MOA emission and sea salt emission. We have shown the MOA emission in Figure 1. Following the reviewer's comment, we added the SSA emissions in fine (Aitken plus accumulation) and coarse modes in Figure S1.

"We note that emissions and burdens of sea salt include the contribution from the coarse mode, which dominates the total sea salt emissions and burdens."

Line 401-406: I find the switch between SSA and sea salt mass rather distracting when discussing the comparison of MOA fractions. Consider making this consistent as to my understanding, SSA is MOA+Sea salt and so it is just a different way of comparing the same values (e.g. MOA/(MOA+Sea salt) or MOA/Sea salt).

We thank the reviewer for the suggestion. We revised the manuscript to make it consistent:

"In B14, the ratio of MOA to sea salt mass burdens reaches up to 2.3 and 1.0 for the Aitken and accumulation modes, respectively. Number concentrations of accumulation mode aerosols near the surface are increased by up to 50% over some regions of the Southern Ocean and Arctic."

Table 5: Perhaps it would be worthwhile to also show the change in mean hygroscopicity of the emitted aerosols as the overall burdens do not seem to change much. Considering some of my comments on section 2.2.2, it would also be worthwhile to see the change in mean number and size of the sea salt and MOA aerosols emitted. I know this greatly varies by region but it would make it clearer to see how adding MOA to the model impacts the number of potential available particles to act as CCN. Also, as previously mentioned, based on the literature discussion, why is the fraction of MOA (MOA/Sea Salt) emission so much lower than reported values of around a few percent for particles larger than 1 micron and even higher for smaller particles?

We thank the reviewer for the suggestion. We are sorry for the confusion and again the numbers for sea salt in Table 5 include the contribution from the coarse mode, while MOA is only contained in the Aitken and accumulation modes. Sea salt burdens in the Aitken and accumulation modes are 0.0014 and 0.48 Tg, respectively, comparing to 8.32 Tg in the coarse mode. We did not output the hygroscopicity, but we have output of number concentrations of aerosols in each mode. Figure R2 shows the change of number concentrations of aerosols in the Aitken and accumulation modes when adding MOA by comparing two model simulations with and without MOA emissions. We notice an increase by up to 50% in the Accumulation mode number concentrations over some regions of the Southern Ocean and Arctic after adding MOA into the model (Figure R2).

[Figure]

Figure R2. Annual mean global distribution of the change $((N_{B14\_D15}-N_{BASE})/N_{BASE})$ of number concentrations of aerosols in the Aitken and accumulation modes, calculated from B15_D15 and BASE experiments.

Line 462: Consider revising to state that impact on clouds via CCN will be discussed next as the INP section follows section 3.2

Thanks. The revised sentence reads as

"Next we will study the MOA effects on clouds via acting as CCN (section 3.2) and INPs (section 3.3), based on model experiments with the B14 emission (Table 4)."

Figure 3: Is there a reason why such a decrease is observed over the Tibetan plateau? Is there a way to add hatching for regions where the changes between simulations are significant or are these changes significant because they are the averages over the nine years?

We thank the reviewer for the suggestion. The decrease CCN over Tibetan plateau after adding MOA
into the model is not statistically significant. We revised Figure 3 as suggested by adding the hatching
for the regions which pass the significance test (at 95% level), shown below

[Figure]

Line 514-515: I would argue that the N12 parametrization does the best job of predicting the INP
concentrations across the entire temperature range as shown in Figure 4. This is mischaracterized in
these sentences. In fact based on the ability of N12, it would be possible to argue that including MOA
emission is not needed to accurately predict the observed INP concentrations.

See our reply to your general comment above. In summary, the field campaigns used in Figure 4 are
marine aerosol dominant/contained scenario campaigns. Thus, dust should not be expected to be the
dominant INPs as indicated by the N12 scheme which only considers dust. Our previous study (Shi and Liu, 2019) also showed that N12 predicts much higher INPs than D15 and agrees better with the INP observations in the Arctic subject to major influences of dust. However, the host aerosol-climate model was found to significantly underestimate the observed dust concentrations by up to a factor of 10. If there were no underestimation of modeled dust, dust INP concentrations from N12 would be much higher than observations. Therefore, N12 has overall the best performance for the wrong reason.

Line 521-523: Following up in the previous comment, the fact that combining M18 and D15 still under predicts, shows that the MOA addition is not as good as just using the N12 parametrization. And one could argue that when using the entire size distribution for dust nucleation, the majority of INPs would accurately simulated.

We thank the reviewer for the suggestion. We agree with the reviewer that the result from D15+M18 scheme is not a perfect match with observations, particularly at temperatures warmer than -15 °C. This may indicate that model misses the representation of marine biological aerosols, which are effective INPs at these warmer temperatures. We also note that N12 overestimates observed INPs even at these warmer temperatures.

Figure 5: How are the parametrizations drawn for the flight campaigns (e.g. Socrates), where INP measured aboard the aircraft are sometimes collected at different altitudes? Line 540-551 and Figure 6: Why was INPs at -25 C and 950 hPa used for this analysis here? For cloud formation, this seems highly irrelevant as it is rarely -25 C at 950 hPa especially in the regions where MOA is expected to be important (over ice-free regions of the Ocean). In fact, how common is it for MPCs to exist at this height. This seems like an unfair height for showing the importance of MOA as INPs as it is a height where MOA concentrations are extremely high due to being within the boundary layer and at a temperature where the ability of MOA to freeze is essentially maximized based on field measurement techniques used at temps above ~-30 C.

We thank the reviewer for the suggestion. In Figures 4 and 5, the simulated INP concentrations are sampled at the same altitudes as the observation data, in the case of aircraft observations.

Figure 6 shows the diagnostic INP at -25℃ at 950 hPa. We used model MOA and dust concentrations at 950 hPa as inputs, and diagnosed the INP concentrations at -25℃. This is similar to what the CFDC-based and filter-based methods measure INPs at a given temperature (often at -25℃). As shown in Figure 6d, the importance of MOA INPs versus dust INPs is not highly variable within the marine boundary layer. Following the reviewer's comment, we also diagnose INP concentrations at -10 and -35℃. As shown in Figure R3, the diagnostic INP distribution patterns are similar to that at -25℃, although the magnitude of INP concentrations (for MOA and dust) are changed. Thus our conclusion of MOA importance as INPs holds at other temperatures.

[Figure]

Figure R3. Vertical cross sections of ratio of MOA INP concentration to dust INP concentration. INP
concentrations are diagnosed at different temperatures (from –10 to –35℃).

To improve the clarity of Figures 4 and 5, we have added a sentence in the figure caption as follows:

"Simulated INPs data are sampled at the same pressures, longitudes and latitudes as the field
measurements."

Line 552-561: I generally agree with the explanation for the observed differences shown in Fig.6d.
However, how does the model handle the sea ice coverage over the Southern Ocean during Austral
winter? As the sea ice extent should extend as far north as approximately 60 S. Therefore, it important
to know how the model handles the emission of MOA during the austral winter and spring months,
especially as the sea ice extent is prescribed based on climatology (see lines 354-356). Does this mean
that the entire year assumes a constant sea ice extent or does it change based on season. Depending on
how this is handled, it could have large implications for both the INP and CCN distribution due to
MOA emissions.

We thank reviewer for the great suggestion. See our reply to your general comment. In
CESM2-CAM6, the prescribed sea ice extent changes with season, and each month is different.

Figure 7: How does the model produce ice nucleation or even MOA at such high latitudes in the Southern Hemisphere at the surface when the center of the Antarctic ice sheet is ~3000 m (700 hPa)?

We thank the reviewer for the comment. We agree with the reviewer that the mountains and ice sheets in the Antarctic are around 3 km in height, and our model also did not have data over these regions. However, there are still some regions at high latitudes in the Southern Hemisphere, where the surface heights are below 400 m at 80° S south (see Figure R3 below). Considering Figure 7 is the annual zonal mean, which includes the data from the summer season and low surface height regions, the data can be extended to low altitudes at high latitudes. We also note that surface level values are very small in Fig.7a.

[Figure]

Figure R4. Global surface height.

Line 579-581: Mentioning seasonal dependence is quite interesting especially concerning my previous comment about the sea-ice extent. Therefore, I think it would be worthwhile to show how the emission of MOA changes over the Southern Ocean between austral summer and winter and how this influences the freezing rates.

That is a very good question, and we added some plots here. We notice that the MOA nucleation rate is strongly dependent on the MOA mass mixing ratio in mixed-phase clouds. The mass mixing ratio of MOA in mixed-phase cloud regions is related to the MOA emission, general circulation (transport), and wet removal (by precipitation). Even though MOA has a smaller emission rate during the austral winter, the effective transport and ice nucleation enhances the ice nucleation rate of MOA in the mixed-phase clouds.

[Figure]

Figure R5. Latitude-pressure cross-sections of annual mean MOA ice nucleation rate in the austral
summer (left) and winter (right).

[Figure]

Figure R6. Latitude-pressure cross-section in the austral summer (left) and winter (right).

Line 590-592: What does this increase in percent mean? Can you report the change in the number of
CDNUMC? Also the fact that there is a difference between austral summer and winter might point to
a change in the sea ice extent in the model as well as biological activity.

We thank the reviewer for the suggestion. To increase the clarity, we revised the sentence in section
3.4 to report the change in the number of CDNUMC and also point to a change in the sea ice extent as

"The vertically-integrated cloud droplet number concentration (CDNUMC) increases by $7.5\times10^4$ cm$^{-2}$
(5.25% in percent change) on the global annual mean, and by $1.1\times10^4$ cm$^{-2}$ (0.94%) and $3.2\times10^5$ cm$^{-2}$
(16.89%) over 20–90°S during the austral winter (June-July-August) and summer
(December-January-February), respectively, by comparing B14_D15_M18 with CTL. This reflects a
strong seasonal variation of MOA emissions due to changes in the sea ice extent as well as biological
activity."

Line 599: why do you switch to an isotherm of -15 now? Previously the -25 isotherm was used and the figures also have the -20 isotherm. Perhaps it is better to be consistent and choose one isotherm throughout or at least explain why different isotherms are chosen.

We thank the reviewer for the comment. We selected the –25°C isotherm level in Figure 6 for a consistent comparison with previous studies (e.g., McCluskey et al., 2019). The –15°C isotherm level was selected in Figure 8 to better represent the mixed-phase cloud feature. –15°C is the most effective temperature for the WBF process, and at this temperature the mixed-phase cloud properties show larger changes than –25°C after introducing more INPs.

We have revised the caption of Figure 8 as follows to explain why we choose another isotherm:

"Figure 8. Annual zonal-mean distributions of (a) surface CCN concentration at S=0.1%, (b) cloud ice number concentration on T=–15°C isotherm, (c) vertically-integrated cloud droplet number concentration, (d) cloud ice mass mixing ratio on T= –15°C isotherm, (e) liquid water path over ocean, (f) ice water path, (g) shortwave cloud forcing, and (h) longwave cloud forcing for CTL (black), B14_D15 (orange), and B14_D15_M18 (green), along with available observations (gray dashed lines) as references. The -15°C isotherm level was used in (b) and (d) to show stronger changes in the mixed-phase cloud properties than the -25°C isotherm."

Line 628-630: Could this also be due to sea ice extent?

We thank the reviewer for the suggestion. We agree with the reviewer and revised the sentence as

  "We also notice that CCN, CDNUMC, and SWCF show smaller changes during the austral winter due to weaker oceanic biological activity and larger sea ice extent."

Line 655-661: Discussion on the differences between bubble bursting (which is implemented in the model) and jet drops (which is not) does not seem necessary. Perhaps it is just fine to just mention that more observations/ fundamental understanding are needed for implementation of jet drops as is not clear why the differences in size of the emitted aerosols matter here. If this is important, please expand on why that could make a significant difference to the observed results and overall importance of MOA.

We thank the reviewer for the suggestion. We explained why jet drops could be potentially important by adding to the discussion:

"These large aerosol particles from jet drops are more effective as CCN and INPs."

**Editorial comments:**

Line 47: "replace to" with "and" as the sentence should read: "temperature is between -38 and 0..."

Done. Thanks.

Line 49: Please consider rephrasing the sentence to read: "INPs have different characteristics
depending on their composition and origin" as it does not make sense as it is written.

Thanks. We have revised the sentence to read as

"INPs have different characteristics depending on their composition and origin."

Line 52: Consider adding some citations when you mention the uncertainty in the ability of black
carbon to act as INPs. To my understanding, the evidence is mounting that BC is irrelevant in the
MPC region.

Thanks. We agree with the reviewer that there is mounting evidence that BC is irrelevant in the
mixed-phase cloud regime (Adams et al., 2020; Kanji et al., 2020; Schill et al., 2020). So we removed
this sentence in the revised manuscript:

"However, large uncertainties exist surrounding the ice nucleating properties of black carbon and
organic carbon from biomass burning and fossil fuel combustion."

Line 74: Please add Ault et al, 2013 to the reference list.

Thanks. We have added Ault et al., 2013 to the reference list.

Line 123: Please add "method" or something similar after: "[Chl-a]-based"

Thanks. We added "method" after "[Chl-a]-based".

Figure 1: Fix unit for micro

Thanks. We have fixed the unit in Figure 1.

Figure 3: The 0.1 % supersaturation is not showing up well. Also, the longitude representation (248 W)
seems a bit odd, but perhaps that's just a personal preference. In the Figure caption, it might be nice to
state that percent change in surface CCN comes from comparing B14-BASE.

We thank the reviewer for the suggestion. We have fixed the issues with Figure 3. We revised the
figure caption of Figure 3 as following:

"Figure 3. Spatial distribution of annual mean percentage changes of surface CCN concentrations at
0.1% supersaturation due to MOA (by comparing B14_D15 and BASE), and vertical distribution of
CCN concentrations at 0.1% supersaturation from eight measurements (solid gray lines), BASE (solid
orange line) and B14_D15 (solid green line)..."

Figure 4: Consider flipping the color bar so that the warmest temperatures are at the top and the
coldest at the bottom.

Thanks. We have changed the color bar as suggested.

674 Figure 6: Please fix panel d to be consistent with the other panels.

Thanks. We have fixed d. We also note that panel d is a different type of figure from other panels but
we think it is better to put it together with other panels in Figure 6 than itself stand-along as a separate
figure. The revised Figure 6 looks

[revised manuscript text omitted]

---

## Author Response (AR2)

**Response to Dr. Manish Shrivastava**

We thank Dr. Manish Shrivastava for handling the review process of our manuscript and his careful reading and review of our manuscript. Our detailed responses to his comments follow. The comments are in blue, our responses are in black, and our corresponding revisions in the manuscript are in red.

**General comments:**

Please clarify the specification of size range of emissions for MOA/Sea salt in primary carbon versus accumulation mode and how it is reported based on model calculations in MAM.

We thank the editor for the comments, we have revised the Table 1 following your comment. We have added the explanations for size ranges in Table 1, and the revised caption for Table 1 reads:

Table 1. Aerosol species in MAM4 modes

| | Accumulation | Aitken | Coarse | Primary Carbon |
|---|---|---|---|---|
| Species[1] | num_a1, so4_a1, pom_a1, soa_a1, bc_a1, dst_a1, ncl_a1, moa_a1 | num_a2, so4_a2, soa_a2, ncl_a2, dst_a2, moa_a2 | num_a3, dst_a3, ncl_a3, so4_a3 | num_a4, pom_a4, bc_a4, (moa_a4 if externally added) |
| Size range[2] | 0.08 – 1 μm | 0.02 – 0.08 μm | 1–10 μm | 0.08 - 1 μm |
| Standard Deviation $\sigma g$ | 1.6 | 1.6 | 1.2 | 1.6 |
| Number-median diameter $Dgn$ | $1.1 \times 10^{-7}$ | $2.6 \times 10^{-8}$ | $2.0 \times 10^{-6}$ | $5.0 \times 10^{-8}$ |
| Low bound $Dgn$ | $5.35 \times 10^{-8}$ | $8.7 \times 10^{-9}$ | $4.0 \times 10^{-7}$ | $1.0 \times 10^{-8}$ |
| High bound $Dgn$ | $4.8 \times 10^{-7}$ | $5.2 \times 10^{-8}$ | $4.0 \times 10^{-5}$ | $1.0 \times 10^{-7}$ |

[1]so4_aX: sulfate mass mixing ratio in mode X; pom_aX: particulate organic matter (POM) mass mixing ratio in mode X; soa_aX: secondary organic aerosol (SOA) mass mixing ratio in mode X; bc_aX: black carbon (BC) mass mixing ratio in mode X; dst_aX: dust mass mixing ratio in mode X; ncl_aX: sea salt mass mixing ratio in mode X; moa_aX: marine organic aerosol (MOA) mass mixing ratio in mode X; and num_aX: number mixing ratio of mode X. *_a1: accumulation mode; *_a2: Aitken mode; *_a3: coarse mode; and *_a4: coarse mode.

[2]The size ranges are only used for sea salt and MOA emissions. MOA emitted in the size range of 0.08-1 $\mu$m is assigned to the primary carbon mode or accumulation mode, depending on the mixing state of MOA with sea salt (Burrows et al., 2018).